# Rab10 inactivation promotes AMPAR trafficking and spine enlargement during long-term potentiation

Jie Wang[1,2†], Jun Nishiyama[2‡], Paula Parra-Bueno[2], Elwy Okaz[2§], Goksu Oz[2], Xiaodan Liu[2], Tetsuya Watabe[2], Irena Suponitsky-Kroyter[2], Timothy E McGraw[3], Erzsebet M Szatmari[2#], Ryohei Yasuda[2]*

[1]Department of Neurobiology, Duke University School of Medicine, Durham, United States; [2]Neuronal Signal Transduction Group, Max Planck Florida Institute for Neuroscience, Jupiter, United States; [3]Weill Cornell Medicine Graduate School of Medical Sciences, New York, United States

*For correspondence: ryohei.yasuda@mpfi.org

Present address: [†]Division of Life Science, The Hong Kong University of Science and Technology, Hong Kong, China; [‡]Program in Neuroscience and Behavioural Disorders, Duke-NUS Medical School, Singapore, Singapore; [§]The Gilbert Family Foundation, Detroit, United States; [#]Department of Physical Therapy, East Carolina University, Greenville, United States

## eLife Assessment

This is an **important** study that describes the development of optical biosensors for various Rab GTPases and explores the contributions of Rab10 and Rab4 to structural and functional plasticity at hippocampal synapses during glutamate uncaging. The evidence supporting the conclusions of the paper is **solid**, and several improvements were noted by the reviewers upon revision, although some persisting inconsistencies would benefit from further clarification.

**Abstract** Rab-dependent membrane trafficking is critical for changing the structure and function of dendritic spines during synaptic plasticity. Here, we developed highly sensitive sensors to monitor Rab protein activity in single dendritic spines undergoing structural long-term potentiation (sLTP) in rodent organotypic hippocampal slices. During sLTP, Rab10 was persistently inactivated (>30 min) in the stimulated spines, whereas Rab4 was transiently activated over ~5 min. Inhibiting or deleting Rab10 enhanced sLTP, electrophysiological LTP, and AMPA receptor (AMPAR) trafficking during sLTP. In contrast, disrupting Rab4 impaired sLTP only in the first few minutes and decreased AMPAR trafficking during sLTP. Thus, our results suggest that Rab10 and Rab4 oppositely regulate AMPAR trafficking during sLTP, and inactivation of Rab10 signaling facilitates the induction of LTP and associated spine structural plasticity.

## Introduction

Structural long-term potentiation (sLTP) is the structural basis of long-term potentiation (LTP) and plays a critical role in learning and memory (*Goto et al., 2021*; *Hayashi-Takagi et al., 2015*; *Matsuzaki et al., 2004*). Upon electrical synaptic stimulation or glutamate uncaging, $Ca^{2+}$ influx through NMDA-type glutamate receptors (NMDARs) into dendritic spines triggers diverse downstream signaling cascades, including $Ca^{2+}$/calmodulin-dependent protein kinase II (CaMKII) and extracellular signal-regulated kinase (ERK), which lead to the increase in spine volume and synaptic efficacy (*Nishiyama and Yasuda, 2015*; *Patterson and Yasuda, 2011*). sLTP has two distinct phases: it starts with a rapid and transient spine enlargement over the first few minutes (transient phase), followed by a sustained enlargement over hours (sustained phase) (*Matsuzaki et al., 2004*; *Murakoshi et al., 2011*). Different pharmacological or genetic manipulations can selectively inhibit these two phases, indicating distinct induction mechanisms (*Murakoshi et al., 2011*). The sustained phase is known to be coupled with

LTP, while the physiological role of the transient phase is less clear (*Murakoshi et al., 2011*; *Matsuzaki et al., 2004*). Nevertheless, the transient phase is associated with the rapid ultrastructural changes of PSD and surrounding membrane and thus may be important for the dramatic synaptic increase in the initial phase of sLTP (*Sun et al., 2021*). In addition, the spine enlargement during the transient phase induces presynaptic potentiation through mechanical force by pushing on the presynaptic bouton (*Ucar et al., 2021*).

During LTP and sLTP, various internal membranes are exocytosed in dendrites in an activity-dependent manner to add spine membrane area, increase surface glutamate receptors, and release plasticity-related peptides (*Harward et al., 2016*; *Lledo et al., 1998*; *Padamsey et al., 2017*; *Park et al., 2004*; *Park et al., 2006*). Particularly, activity-dependent synaptic delivery of GluA1 subunit of AMPA-type glutamate receptor (AMPAR) is considered one of the major mechanisms to increase postsynaptic glutamate sensitivity during LTP (*Shi et al., 1999*; *Sheng and Lee, 2001*; *Malinow and Malenka, 2002*; *Huganir and Nicoll, 2013*; *Hayashi et al., 2000*). Several members of the post-synaptic exocytosis machinery have been identified, including the soluble NSF-attachment protein receptor (SNARE) proteins, complexin, synaptotagmins, and myosin Vb (*Lledo et al., 1998*; *Wu et al., 2017*; *Wang et al., 2008*; *Ahmad et al., 2012*; *Lu et al., 2001*; *Jurado et al., 2013*; *Kennedy et al., 2010*).

Among the molecules regulating intracellular trafficking, Rab GTPases constitute the largest Ras subfamily, with more than 60 members localized to distinct intracellular domains (*Zerial and McBride, 2001*; *Stenmark, 2009*). As small GTPases, Rab proteins switch between two states, the guanosine-5'-triphosphate (GTP)-bound 'active' state and the guanosine diphosphate (GDP)-bound 'inactive' state. Specific guanine nucleotide exchange factors (GEFs) and GTPase-activating proteins (GAPs) regulate the conversion between these two states (*Stenmark, 2009*). Once activated, Rab GTPases recruit diverse downstream effectors to coordinate intracellular transport during vesicle budding, movement, tethering, and fusion (*Hutagalung and Novick, 2011*). Several members of Rab GTPases have been shown to coordinate AMPAR trafficking during synaptic plasticity (*Hausser and Schlett, 2019*). Rab8 and Rab11 regulate the exocytosis of AMPARs during LTP (*Gerges et al., 2004*; *Brown et al., 2007*; *Correia et al., 2008*), whereas Rab5 drives the endocytosis of AMPARs during long-term depression (LTD) (*Brown et al., 2005*). Rab4 is dispensable for LTP but is required for the maintenance of spine morphology (*Correia et al., 2008*). Rab4 also regulates both the fast and slow endosomal recycling to the plasma membrane (*van der Sluijs et al., 1992a*; *Sönnichsen et al., 2000*; *Ehlers, 2000*). However, the function of other Rabs in synaptic plasticity is less known. Notably, Rab10 is expressed in dendrites (*Taylor et al., 2015*; *Liu et al., 2013*; *Wang et al., 2011*; *Zou et al., 2015*) and has been implicated in Alzheimer's disease resilience (*Tavana et al., 2019*; *Ridge et al., 2017*). Thus, Rab10 is potentially involved in membrane trafficking during synaptic plasticity.

In the present study, we developed a general design of FRET/FLIM-based sensors for various Rab proteins. Among these sensors, we further focused on the spatiotemporal dynamics of Rab10 and Rab4 activity during sLTP. We found that postsynaptic stimulation leads to the persistent inactivation of Rab10 and transient activation of Rab4 in spines undergoing sLTP. These Rab activity changes are dependent on NMDAR and CaMKII activation. Moreover, knock-down analyses demonstrate that Rab10 serves as a negative regulator of sLTP, whereas Rab4 contributes to the transient phase of sLTP. In addition, postnatal deletion of Rab10 from excitatory neurons enhanced sLTP and electrophysiological LTP at Schaeffer collateral pathway. Furthermore, Rab10 negatively regulates activity-dependent GluA1 trafficking into the stimulated spines during sLTP, while Rab4 positively controls this process. Therefore, our results suggest that Rab10 inhibits AMPAR trafficking during synaptic potentiation, and NMDAR-dependent inactivation of Rab10 facilitates LTP induction. On the contrary, Rab4 promotes AMPAR trafficking during sLTP and NMDAR-dependent activation of Rab4 regulates the transient phase of sLTP.

## Results

### Highly sensitive and selective FRET sensors for Rab proteins

To image Rab signaling activity in single dendritic spines, we developed FRET sensors for Rab4, 5, 7, 8, and 10. The Rab sensors have two components: (1) the Rab protein tagged with a fluorescent protein that serves as a FRET donor (monomeric enhanced green fluorescent protein (mEGFP)-Rab

or mTurquoise2-Rab) and (2) a Rab binding domain (RBD) from a specific effector protein that is tagged with two fluorescent proteins as FRET acceptors (mCherry-RBD-mCherry or mVenus-RBD-mVenus). When the Rab protein is activated, RBD and Rab increase their binding and thus increase FRET between the donor and acceptor fluorophores (*Figure 1a*). The fraction of Rab bound to the RBD (binding fraction) was calculated by measuring the fluorescence lifetime of the donor (*Yasuda, 2012*; *Yasuda, 2006*).

For Rab4, 5, 7, 8, and 10, we used Rabenosyn5 [439-503], EEA1 [36-126], FYCO1 [963–1206], Rim2 [27-175], and Rim1 [20-227] as RBDs, respectively (*Simonsen et al., 1998*; *Pankiv et al., 2010*; *Fukuda, 2003*; *Eathiraj et al., 2005*). To test the sensitivity and specificity of these Rab sensors, we took three approaches. First, we transfected wild-type (WT)-Rab, dominant-negative (DN)-Rab, and constitutively active (CA)-Rab sensors in HEK 293T cells. As expected, DN-Rab and CA-Rab sensors displayed lower and higher binding fractions than the WT-Rab sensor, respectively, indicating the sensitivity of Rab sensors (*Figure 1b and c*). Particularly, Rab4 and Rab10 sensors showed significantly different binding fractions between WT- and DN- or CA-Rabs (*Figure 1b and c*). For the Rab10 sensor, we used the mTurquoise2-mVenus pair instead of the mEGFP-mCherry pair since it reported higher binding fraction differences between WT- and DN-, as well as WT- and CA-Rab10 sensors in HEK 293T cells (*Figure 1b,c*, *Figure 1—figure supplement 1a-d*; *Takahashi et al., 2015*; *Goedhart et al., 2012*). Second, we coexpressed a Rab sensor with the corresponding Rab GAPs or GEFs to test whether the sensor could respond to the known upstream signaling (*Yoshimura et al., 2010*; *Tall et al., 2001*; *Rink et al., 2005*; *Itoh et al., 2006*; *Hattula et al., 2002*; *Fukuda et al., 2008*; *Chamberlain et al., 2004*). Indeed, these GAPs and GEFs respectively decreased and increased the activity of Rab proteins as reported by the sensor (*Figure 1d–l*). Compared with Rab5 and Rab8, Rab4, Rab7, and Rab10 sensors displayed higher binding fraction changes in response to GAPs or GEFs (*Figure 1d–i*). Third, we measured Rab activity change in response to N-Methyl-D-aspartic acid (NMDA) application in neurons (*Murakoshi et al., 2011*). We biolistically transfected rat organotypic hippocampal slices with each Rab sensor and imaged the proximal apical dendrites of CA1 pyramidal neurons. Bath application of NMDA (15 µM, 2 min) in zero extracellular $Mg^{2+}$ triggered a robust activation of Rab sensors in the dendrites, suggesting that Rab sensors could report neuronal Rab activities (*Figure 1j*). Nevertheless, these Rab sensors showed different activation kinetics upon NMDA stimulation (*Figure 1j*, *Figure 1—figure supplement 1e*). Rab4 sensor was rapidly activated, peaked at 3 min and subsequently decayed (*Figure 1j*, *Figure 1—figure supplement 1e*). On the contrary, Rab5, 7, 8, and 10 sensors displayed a gradually accumulated activation pattern, which peaked at 6–11 min and decreased afterwards (*Figure 1j*, *Figure 1—figure supplement 1e*). Notably, the Rab10 sensor was transiently inactivated in the first 2 min, possibly reflecting a fast response to NMDAR activation (*Figure 1j*, *Figure 1—figure supplement 1e*). Overall, these results demonstrate that our Rab sensor strategy is generalizable to many Rab proteins.

## Rab10 is persistently inactivated in the stimulated spines during sLTP

We biolistically transfected cultured organotypic hippocampal slices of rats with Rab10 sensor and imaged the secondary apical dendrites of CA1 pyramidal neurons. mTurquoise2-Rab10, the sensor donor, was continuously distributed in the dendrites and spines, without showing an endosomal punctate pattern (*Figure 2a*). To eliminate the effect of protein overexpression on its localization, we probed endogenous Rab10 by CRISPR-Cas9-mediated homology-directed repair (SLENDR) technique in vivo (*Figure 1—figure supplement 1f–j*; *Mikuni et al., 2016*). Endogenous Rab10 displayed a heterogeneous pattern: it is widely distributed along the dendrite and protrudes into the spines (*Figure 1—figure supplement 1i*). To elucidate the subcellular compartmentalization of Rab10, we further examined its colocalization with exogenously expressed endosomal markers in vivo (*Figure 1—figure supplement 1j*). Rab10 is significantly overlapped with Rab7-labeled lysosomes, partially colocalized with Rab11-labeled recycling endosomes, but separated from Rab5-labeled early endosomes (*Figure 1—figure supplement 1j*).

To characterize the spatiotemporal dynamics of Rab10 activity in single dendritic spines during sLTP, we combined two-photon glutamate uncaging with 2pFLIM (*Murakoshi et al., 2011*; *Harward et al., 2016*; *Harvey et al., 2008*; *Hedrick et al., 2016*), and imaged the secondary apical dendrites of CA1 pyramidal neurons expressing the Rab10 sensor at 25–27°C. Upon focal glutamate uncaging (0.5 Hz, 60 s) in zero extracellular $Mg^{2+}$, a single spine underwent a rapid volume increase within

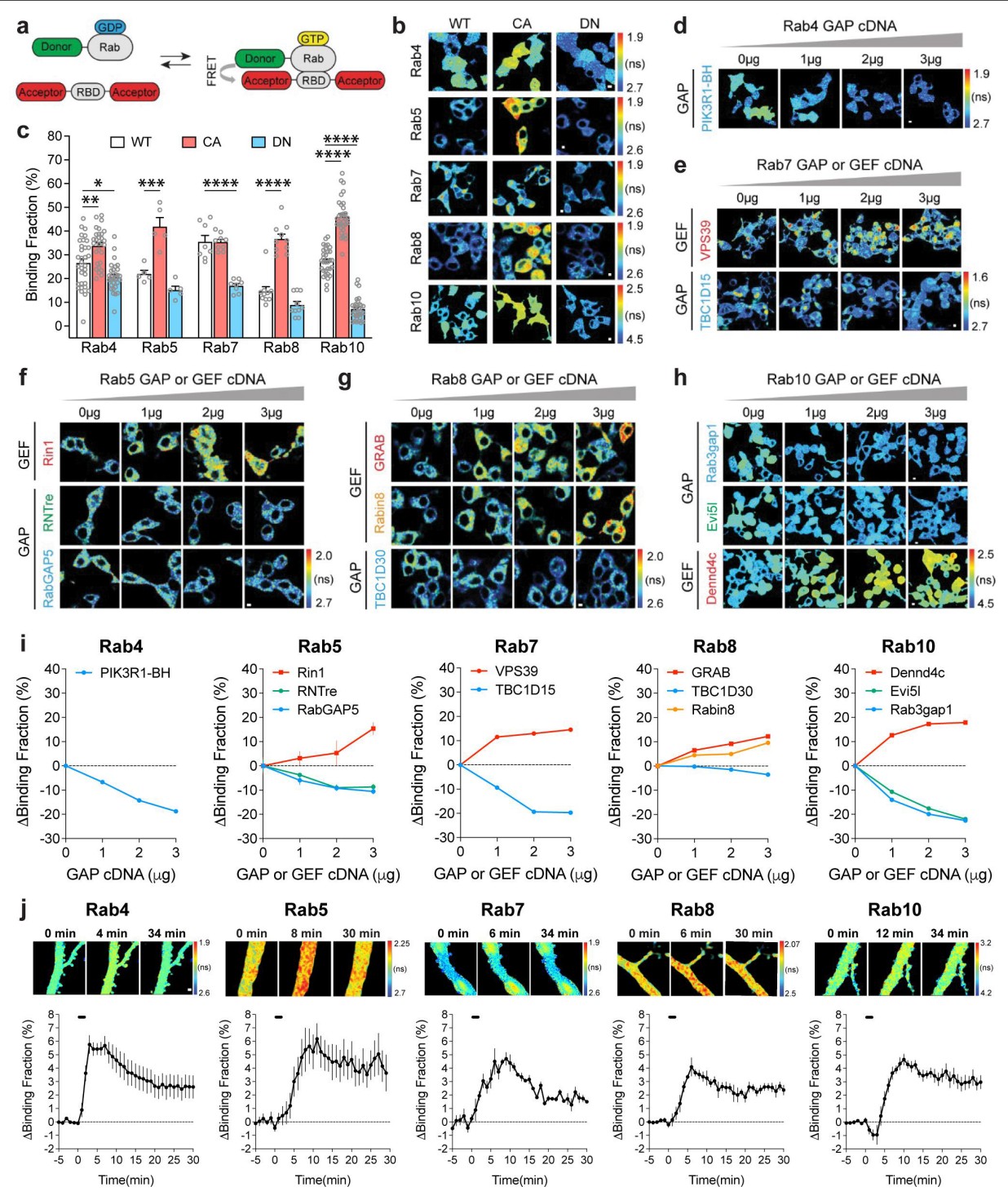

**Figure 1.** Design and characterization of Rab FRET sensors. (**a**) Schematic diagrams illustrating the FRET sensor design for Rab proteins. RBD: Rab binding domain; GDP: guanosine diphosphate; GTP: guanosine-5'-triphosphate. (**b**) Representative fluorescence lifetime images of HEK 293T cells transfected with Rab wild type (WT), dominant negative (DN, Rab4a [S27N], Rab5a [S34N], Rab7a [T22N], Rab8a [S22N] and Rab10 [T23N]) and constitutively active (CA, Rab4a [Q72L], Rab5a [Q79L], Rab7a [Q67L], Rab8a [Q67L] and Rab10 [Q68L]) sensors. FRET donor/acceptor pair is mEGFP/mCherry for Rab4, 5, 7, and 8; for Rab10, FRET donor/acceptor pair is mTurquoise2/mVenus. Warmer color indicates shorter lifetime and higher activity. Scale bars represent 5 μm. (**c**) Quantification of binding fraction for experiments in b. Data represent mean ± SEM. N=5–33. One-way ANOVA followed by Bonferroni's multiple comparison tests was performed (* p<0.05, ** p<0.01, *** p<0.001, **** p<0.0001). (**d-h**) Representative fluorescence lifetime images of individual Rab wild type sensor cotransfected with the corresponding GAP (green and blue) or GEF (red and orange) cDNAs. For GEF, Rin1,

*Figure 1 continued on next page*

*Figure 1 continued*

VPS39, GRAB, Dennd4c are used for Rab5, Rab7, Rab8, and Rab10. For GAP, we used breakpoint cluster region homology (BH) domain of PIK3R1 (PIK3R1-BH or PIK3R1[114-313]) for Rab4 GAP, RNTre and RabGAP5 for Rab5 GAP, TBC1D15 for Rab7, TBC1D30 and Rabin8 for Rab8, and Evi5l and Rab3gap1 for Rab10. Scale bars represent 5 µm. (**i**) Quantification of binding fraction change for experiments in **d-h**. N=3–20. Data represent mean ± SEM. (**j**) Activation of Rab sensors by NMDA application in rat hippocampal CA1 pyramidal neurons. Upper: representative fluorescence lifetime images of Rab sensor-expressing neurons in response to bath application of 15 µM NMDA for 2 min. Scale bar is 1 µm. Lower: quantification of binding fraction change for upper lane experiments. Data represent mean ± SEM. N=4–11 for each group.

The online version of this article includes the following source data and figure supplement(s) for figure 1:

**Figure supplement 1.** mTurquoise2-Rab4 and mEGFP-Rab10 FRET sensors in HEK 293T Cells, all Rab sensor activity changes upon NMDA application, and localization of endogenous Rab10.

**Figure supplement 1—source data 1.** Original gel photos.

**Figure supplement 1—source data 2.** Original gel photos with annotations.

the first few minutes ($\Delta V_{transient}$ = 275.9 ± 25.0%; *Figure 2a, e and f*), which decayed to a smaller but sustained volume increase lasting more than 30 min ($\Delta V_{sustained}$ = 79.6 ± 6.7%; *Figure 2a, e and g*), consistent with the previous studies (*Harvey et al., 2008*; *Hedrick et al., 2016*; *Matsuzaki et al., 2004*; *Murakoshi et al., 2011*; *Harward et al., 2016*). Concomitant with the spine enlargement, Rab10 showed a rapid and spine-specific decrease of activity in the stimulated spines, which lasted for more than 30 min (*Figure 2a–d*). Neither the changes in spine volume, basal binding fraction, nor the changes in the binding fraction of Rab10 sensor correlated with the initial spine size (*Figure 2—figure supplement 1a–e*). To verify the specificity of the sensor response, we replaced the RBD of the Rab10 sensor with the RBD of Rab4 sensor (Rabenosyn5 [439-503], false acceptor). This false acceptor sensor did not show a significant change in FRET (measured as the binding fraction change) in the stimulated spines (*Figure 2c,d,f,g*, *Figure 2—figure supplement 2*). In addition, the Rab10 sensor displayed a similar inactivation pattern at a near-physiological temperature (33–35°C; *Figure 2—figure supplement 3a–d*). The inactivation of Rab10 was correlated neither with the concentration of donor nor with that of the acceptor, suggesting that overexpressing Rab10 sensor has minimal effects on Rab10 inactivation (*Figure 2—figure supplement 3e–h*). In summary, our results demonstrate that Rab10 is persistently inactivated in the stimulated spines during sLTP, and this inactivation is compartmentalized in the stimulated spines.

The persistent inactivation of Rab10 during sLTP may be caused by the dilution of active Rab10 proteins due to the rapid spine enlargement. To examine this possibility, we compared the basal binding fraction of Rab10 sensor in spines and dendrites and analyzed the subset of spines with lower Rab10 basal activity than the dendrite (*Figure 2—figure supplement 4b–d*). We found that Rab10 activity still decreased against the gradient at the spine neck upon sLTP induction (*Figure 2*, *Figure 2—figure supplement 4d*). In addition, the binding fraction of the Rab10 sensor is independent of donor intensity (*Figure 2—figure supplement 4a*). Thus, instead of passive signal dilution, the persistent inactivation of Rab10 requires active biological processes, presumably by translocating Rab10-positive vesicles out of the spine or directly inactivating Rab10 in the spine. To examine whether Rab10 distribution is changed during sLTP, we further analyzed the intensity of Rab10 donor in the stimulated spines and dendrites. Upon sLTP induction, Rab10 donor intensity significantly increased in the stimulated spines but remained unchanged in the dendrites (*Figure 2—figure supplement 5a*). The dramatic increase of Rab10 intensity in the stimulated spines was probably due to passive diffusion by spine enlargement. Therefore, we further analyzed the mean intensities of Rab10 donor in the stimulated spines and dendrites, which were similar prior to and post sLTP induction (*Figure 2—figure supplement 5b and c*).

To further identify signaling pathways that inactivate Rab10 during sLTP, we applied pharmacological inhibitors targeting potential upstream components (*Patterson and Yasuda, 2011*; *Hedrick et al., 2016*). Inhibition of NMDARs by 2-amino-5-phosphonopentanoic acid (AP5, 50 µM) completely abolished Rab10 inactivation and spine enlargement (*Figure 2c,d,f,g*, *Figure 2—figure supplement 6a,b*). Application of CN21 (10 µM), a CaMKII inhibitory peptide (*Chang et al., 1998*; *Vest et al., 2007*), abolished Rab10 inactivation while partially attenuating the volume change (*Figure 2c,d,f,g*, *Figure 2—figure supplement 6c,d*). In contrast, mitogen-activated protein kinase (MAPK)/ERK kinase (MEK) inhibitor U0126 (20 µM) had no effect on Rab10 inactivation, although it impaired sLTP during the sustained phase (*Figure 2c,d,f,g*, *Figure 2—figure supplement 6e,f*). These results demonstrate

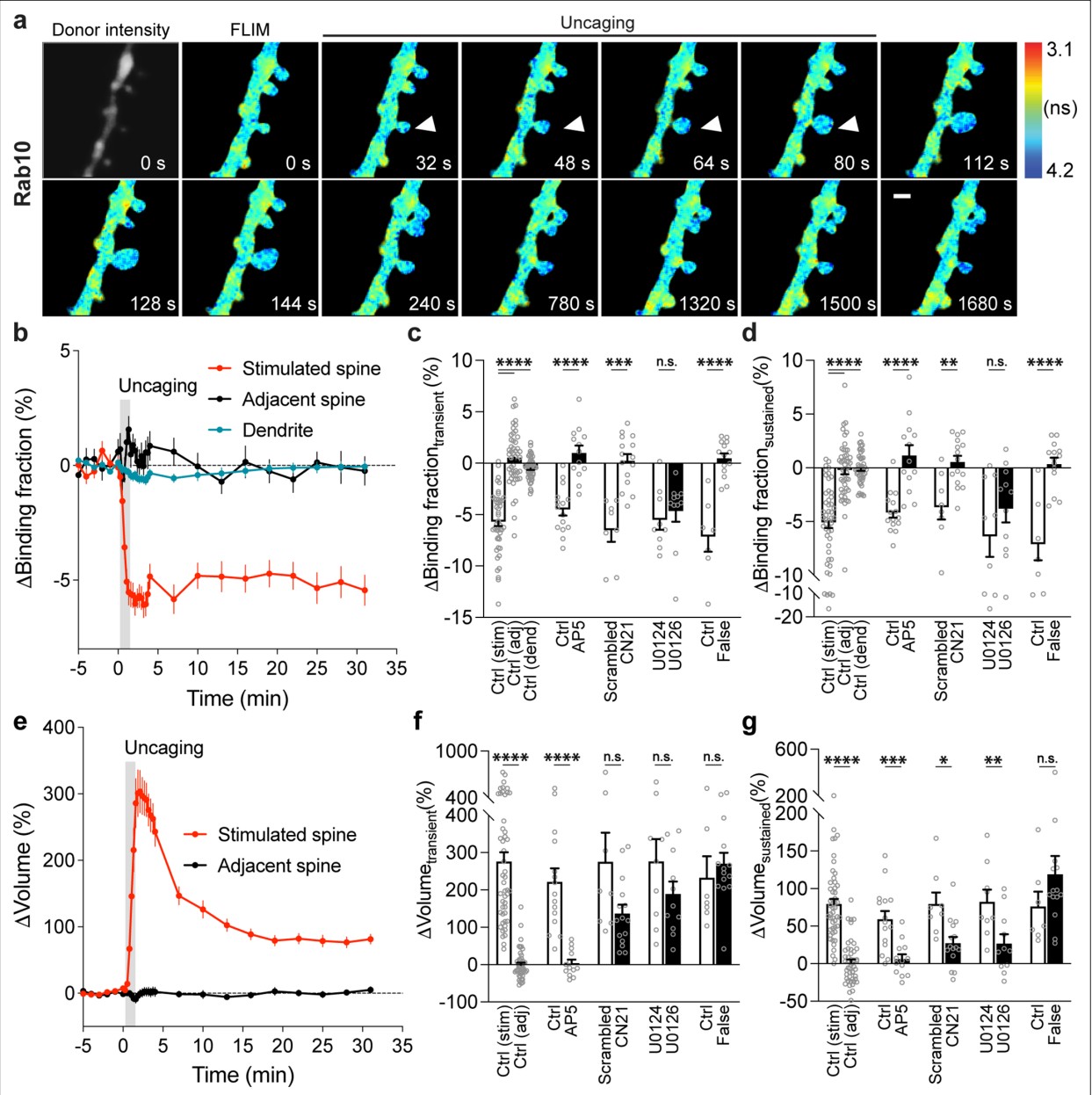

**Figure 2.** Spatiotemporal dynamics of Rab10 inactivation during sLTP in single spines. (**a**) Representative donor fluorescence intensity image (gray) and fluorescence lifetime (FLIM, colorful) images of Rab10 inactivation during sLTP induced by two-photon glutamate uncaging. Arrowheads indicate the stimulated spine. The colder color indicates longer lifetime and lower Rab10 activity. Scale bar represents 1 μm. (**b**) Averaged time courses of Rab10 inactivation measured as binding fraction changes between mTurquoise2-Rab10 and mVenus-Rim1 [20-277]-mVenus in the stimulated spine (red), adjacent spine (black), and dendrite (blue). Gray rectangle bar indicates glutamate uncaging (0.5 Hz, 60 s). Data are presented in mean ± SEM. N=49/42 (spine/neuron), 49/42 (spine/neuron) and 49/42 (dendrite/neuron) for the stimulated spine, adjacent spine, and dendrite, respectively. (**c,d**) Quantification of Rab10 binding fraction changes in the transient phase (c, averaged over 1.3–4 min) and sustained phase (**d**, averaged over 19–31 min) in the stimulated spine (stim), adjacent spine (adj), and dendrite (dend) for the same experiments as b. Data represent mean ± SEM. One-way ANOVA followed by Bonferroni's multiple comparison tests was performed for the stimulated spine, adjacent spine, and dendrite (**** p<0.0001). Effects of pharmacological agents on Rab10 inactivation in the stimulated spines are also presented. All pharmacological inhibition experiments were paired with controls from the same batch of slices. Data represent mean ± SEM. Student's t-tests were performed (n.s., not significant, ** p<0.01, *** p<0.001, **** p<0.0001). N=15/14, 13/11, 8/7, 15/12, 9/8, and 11/8 (spine/neuron) for Ctrl, AP5, scrambled, CN21, U0124, and U0126, respectively. When mTurquoise2-Rab10 was paired with a false acceptor (False), mVenus-Rabenosyn5 [439-503]-mVenus, little activity change was observed. Data represent mean ± SEM. Student's t-tests were performed (**** p<0.0001). N=7/7 and 14/9 (spine/neuron) for Ctrl and False, respectively. (**e**) Averaged time courses of spine volume change in the same experiment as b. Data represent mean ± SEM. (**f,g**) Quantification of spine volume changes in the transient

*Figure 2 continued on next page*

*Figure 2 continued*

phase (**f**, averaged over 1.3–4 min) and sustained phase (**g**, averaged over 19–31 min) for the same experiments as c and d. Data represent mean ± SEM. Student's t-tests were used for all groups (n.s., not significant, * p<0.05, ** p<0.01, *** p<0.001, **** p<0.0001).

The online version of this article includes the following figure supplement(s) for figure 2:

**Figure supplement 1.** Relationship between initial spine volume and basal Rab GTPase activity, spine volume change, or activity change during sLTP.

**Figure supplement 2.** Binding fraction changes of mTurquoise2-Rab10 paired with false acceptor during sLTP.

**Figure supplement 3.** Inactivation of Rab10 during sLTP induced at near physiological temperature.

**Figure supplement 4.** Properties of basal binding fraction and binding fraction change in the stimulated spines for Rab10 and Rab4 sensors.

**Figure supplement 5.** Intensity and mean intensity of Rab10 and Rab4 sensor donors in the stimulated spines and dendrites during sLTP.

**Figure supplement 6.** Rab10 inactivation under manipulations of putative upstream signaling pathways.

that Rab10 is persistently inactivated in the stimulated spines during sLTP, and this inactivation is dependent on NMDARs and CaMKII but not on the MAPK/ERK signaling pathway.

## Rab4 is transiently activated in the stimulated spines during sLTP

Next, we measured the spatiotemporal profile of Rab4 activity in dendrites during sLTP induced in single spines. Consistent with its localization in early and recycling endosomes (*Ehlers, 2000*; *Sönnichsen et al., 2000*; *van der Sluijs et al., 1992b*), mEGFP-Rab4, the sensor donor, showed a punctate distribution pattern (*Figure 3a*). In the basal state, Rab4 sensor activity is not correlated with the donor intensity and was lower in the spines than dendrites (*Figure 2—figure supplement 4e–g*). Glutamate uncaging (0.5 Hz, 60 s) induced a transient and sustained volume increase in the stimulated spines of neurons expressing Rab4 sensor ($\Delta V_{transient}$ = 341.0 ± 29.4% and $\Delta V_{sustained}$ = 77.6 ± 8.0%; *Figure 3a and e–g*). In contrast to the persistent inactivation of Rab10, Rab4 activity was transiently elevated in the stimulated spines and decayed, with no activity change in the adjacent spines or dendrites (*Figure 3a–d*). This enhanced spine activity exceeds that of dendrite during sLTP, suggesting that the activation requires active processes (*Figure 2—figure supplement 4h*). Similarly to Rab10, we examined the intensity and mean intensity of Rab4 donor in the stimulated spines and dendrites. In the transient phase of sLTP, both the intensity and mean intensity of Rab4 donor significantly increased in the stimulated spines, suggesting recruitment of Rab4 into the stimulated spines (*Figure 2—figure supplement 5d–f*). In contrast, there was no donor intensity or mean intensity change in the dendrites (*Figure 2—figure supplement 5d–f*). Moreover, the initial spine volume was not correlated either with the volume changes, the basal activity of Rab4 reported or the level of Rab4 activation reported by the sensor (*Figure 2—figure supplement 1f–j*). Replacing the RBD of Rab4 sensor with the RBD for Rab10 sensor (Rim1 [20-227], false acceptor) showed no change of the binding fraction in the stimulated spines during sLTP, indicating the specificity of the sensor response (*Figure 3c,d,f,g*, *Figure 3—figure supplement 1*). In addition, the Rab4 sensor showed a similar transient activation pattern at a near-physiological temperature (33–35°C; *Figure 3—figure supplement 2*). The obtained signal was independent of the expression level of Rab4 sensor (*Figure 3—figure supplement 2e–h*). Thus, our results demonstrate that Rab4 is transiently activated in the stimulated spines during sLTP, and this activation is compartmentalized in the stimulated spines.

To identify signaling pathways that activate Rab4 during sLTP, we applied pharmacological inhibitors targeting putative upstream components. Inhibition of NMDARs by AP5 (50 µM) completely abolished Rab4 activation and spine enlargement (*Figure 3c,d,f,g*, *Figure 3—figure supplement 3a,b*). Application of CN21 (10 µM) decreased Rab4 activity and volume changes both in the transient phase and sustained phase (*Figure 3c,d,f,g*, *Figure 3—figure supplement 3c,d*). In contrast, MEK inhibitor U0126 (20 µM) had no effect on Rab4 activation, although it impaired sLTP during the sustained phase (*Figure 3c,d,f,g*, *Figure 3—figure supplement 3e,f*). Altogether, Rab4 activation during sLTP is dependent on NMDARs and CaMKII, but not on the MAPK/ERK signaling pathway.

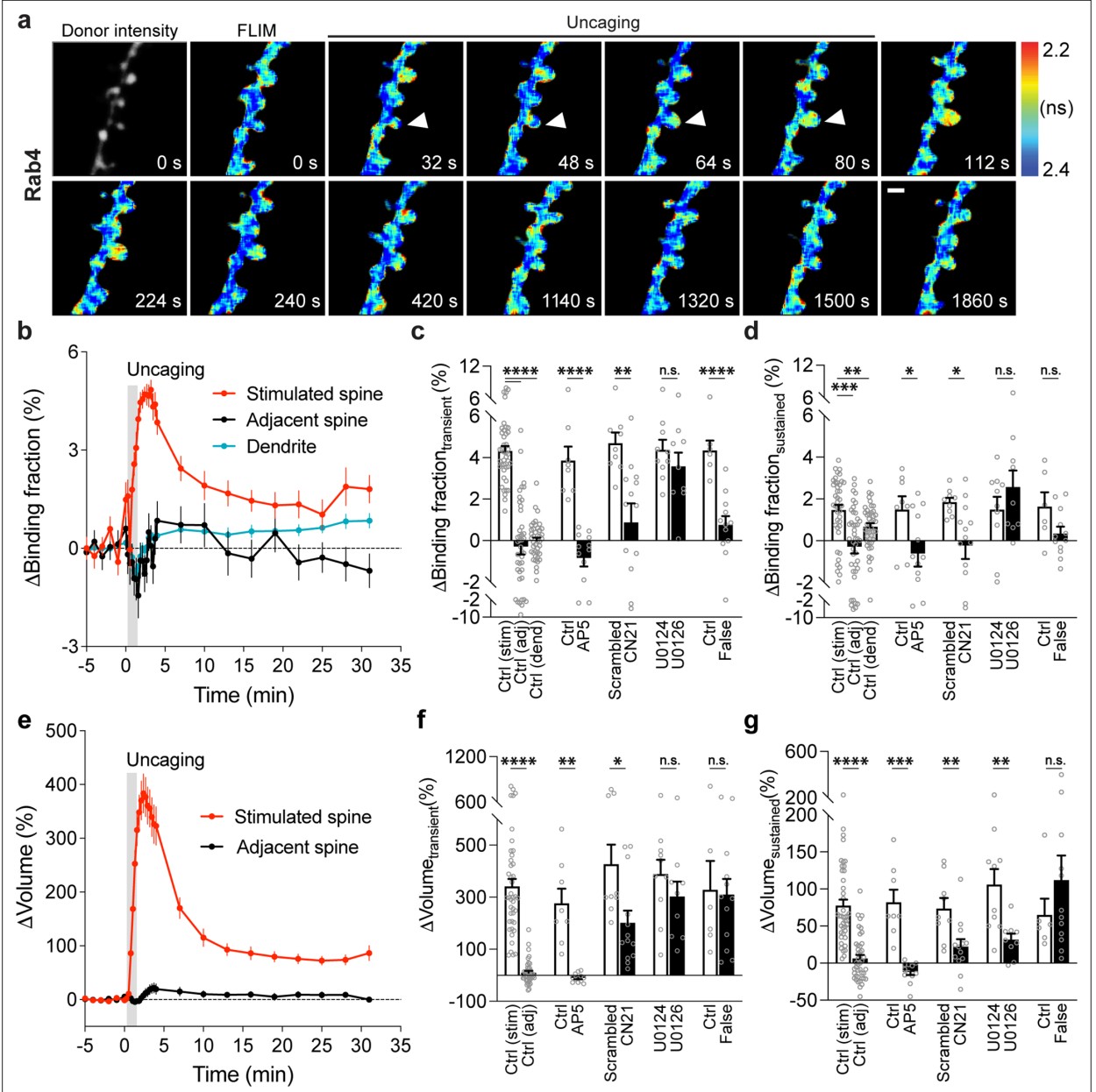

**Figure 3.** Spatiotemporal dynamics of Rab4 activation during sLTP induced at single spines. (**a**) Representative donor fluorescence intensity image (gray) and fluorescence lifetime (FLIM, colorful) images of Rab4 activation during sLTP induced by two-photon glutamate uncaging. Arrowheads indicate the stimulated spine. The warmer color indicates shorter lifetime and higher Rab4 activity. Scale bar represents 1 μm. (**b**) Averaged time courses of Rab4 activation measured as binding fraction changes between mEGFP-Rab4 and mCherry-Rabenosyn5 [439-503]-mCherry in the stimulated spine (red), adjacent spine (black), and dendrite (blue). Gray rectangle bar indicates glutamate uncaging (0.5 Hz, 60 s). Data are presented in mean ± SEM. N=42/34 (spine/neuron), 42/34 (spine/neuron), and 42/34 (dendrite/neuron) for the stimulated spine, adjacent spine, and dendrite, respectively. (**c, d**) Quantification of Rab4 binding fraction changes in the transient phase (**c**, averaged over 1.3–4 min) and sustained phase (**d**, averaged over 19–31 min) in the stimulated spine (stim), adjacent spine (adj), and dendrite (dend) for the same experiments as b. Data represent mean ± SEM. One-way ANOVA followed by Bonferroni's multiple comparison tests was used for the stimulated spine, adjacent spine, and dendrite (** p<0.01, *** p<0.001, **** p<0.0001). Effects of pharmacological agents on Rab4 activation in the stimulated spines are also presented. All pharmacological inhibition experiments were paired with controls from the same batch of slices. Data represent mean ± SEM. Student's t-tests were performed (n.s., not significant, * p<0.05, ** p<0.01, **** p<0.0001). N=8/6, 12/9, 9/8, 13/11, 10/8, and 9/8 (spine/neuron) for Ctrl, AP5, scrambled, CN21, U0124, and U0126, respectively. When mEGFP-Rab4 was paired with a false acceptor (False), mCherry-Rim1 [20-227]-mCherry, little activity change was observed in the stimulated spines. Data represent mean ± SEM. Student's t-tests were performed (n.s., not significant, **** p<0.0001). N=6/5 and 12/9 (spine/neuron) for Ctrl and False, respectively. (**e**) Averaged time courses of spine volume change in the same experiments as b. Data represent mean ± SEM. (**f, g**) Quantification of spine

*Figure 3 continued on next page*

**Figure 3 continued**

volume changes in the transient phase (**f**, averaged over 1.3–4 min) and sustained phase (g, averaged over 19–31 min) for the same experiments as **c** and **d**. Data represent mean ± SEM. Student's t-tests were used for all groups (n.s., not significant, * p<0.05, ** p<0.01, *** p<0.001, **** p<0.0001).

The online version of this article includes the following figure supplement(s) for figure 3:

**Figure supplement 1.** Changes in the binding fraction of mEGFP-Rab4 paired with false acceptor during sLTP.

**Figure supplement 2.** Activation of Rab4 during sLTP induction at near physiological temperature.

**Figure supplement 3.** Pharmacological evaluations of upstream signaling pathways mediating Rab4 activation.

## Disruption of Rab10 signaling enhances structural and electrophysiological LTP, whereas disruption of Rab4 signaling inhibits the transient phase of structural LTP

Given that Rab10 and Rab4 display opposing activity profiles during sLTP, we hypothesized that they would have opposite functions in spine structural plasticity. To test this hypothesis, we knocked down Rab10 or Rab4 by respective shRNA and examined the effects of these manipulations on sLTP. We biolistically transfected cultured organotypic hippocampal slices of rats with scrambled shRNA control or shRNA against *Rab10* or *Rab4*, together with mEGFP as the volume marker. Compared with scrambled shRNA control, knockdown of Rab10 had no effect on spine size but increased spine density (*Figure 4a*, *Figure 4—figure supplement 1a,b,f,g,h,i*). Knockdown of Rab4 had no effect on spine size or density (*Figure 4a*, *Figure 4—figure supplement 1a,b,e,g,h,i*).

We further induced sLTP by two-photon glutamate uncaging in single spines of neurons expressing mEGFP (*Murakoshi et al., 2011*; *Lee et al., 2009*). Under control conditions with scrambled shRNA, application of a train of glutamate uncaging pulses (0.5 Hz, 60 s) in zero extracellular $Mg^{2+}$ induced a rapid spine volume increase in the transient phase, which decayed to a sustained enlarged volume for more than 30 min (*Figure 4a and c*). However, knockdown of Rab10 by shRNA enhanced spine enlargement both in the transient and sustained phase of sLTP, which was rescued by co-expressing shRNA-resistant Rab10 (for scrambled shRNA, $\Delta V_{transient} = 215.5 \pm 16.6\%$ and $\Delta V_{sustained} = 64.4 \pm 9.9\%$; for *Rab10* shRNA, $\Delta V_{transient} = 309.0 \pm 36.3\%$ and $\Delta V_{sustained} = 140.6 \pm 16.4\%$; for Rab10 rescue, $\Delta V_{transient} = 213.5 \pm 19.7\%$ and $\Delta V_{sustained} = 87.4 \pm 11.7\%$; *Figure 4a,c,e*, *Figure 4—figure supplement 1a,b,d,f,g*). In contrast, knockdown of Rab4 by shRNA significantly impaired the transient phase of sLTP while leaving the sustained phase intact (for scrambled shRNA, $\Delta V_{transient} = 291.6 \pm 35.6\%$ and $\Delta V_{sustained} = 76.1 \pm 10.0\%$; for *Rab4a/4b* shRNA, $\Delta V_{transient} = 119.8 \pm 15.4\%$ and $\Delta V_{sustained} = 73.8 \pm 12.1\%$; *Figure 4a,c,e*, *Figure 4—figure supplement 1a,b,e,g*). This phenotype was rescued by coexpression of shRNA-resistant Rab4a ($\Delta V_{transient} = 223.1 \pm 32.8\%$ and $\Delta V_{sustained} = 84.2 \pm 12.3\%$; *Figure 4a,c,e*, *Figure 4—figure supplement 1c*). Overall, these results suggest that Rab10 negatively regulates both the transient and sustained phases of sLTP, while Rab4 is required for the transient phase of sLTP.

As an alternative strategy to inhibit Rab10 and Rab4 functions, we examined the effects of overexpressing DN-Rab mutants on spine structural plasticity. Consistent with the shRNA results, DN-Rab10 enhanced both the transient and sustained phase of sLTP (for control, $\Delta V_{transient} = 209.2 \pm 31.1\%$ and $\Delta V_{sustained} = 64.8 \pm 9.4\%$; for Rab10 DN, $\Delta V_{transient} = 332.0 \pm 31.8\%$ and $\Delta V_{sustained} = 125.0 \pm 17.4\%$; *Figure 4b, d and f*), while DN-Rab4a selectively inhibited the transient phase of sLTP (for control, $\Delta V_{transient} = 357.4 \pm 52.0\%$ and $\Delta V_{sustained} = 83.5 \pm 10.8\%$; for Rab4a DN, $\Delta V_{transient} = 198.1 \pm 32.1\%$ and $\Delta V_{sustained} = 70.8 \pm 11.0\%$; *Figure 4b, d and f*). Moreover, we evaluated the effect of overexpressing CA-Rab10 or CA-Rab4 on sLTP. These manipulations in general caused opposite phenotypes to the DN mutants: CA-Rab10 decreased both the transient and sustained phase of sLTP (for control, $\Delta V_{transient} = 209.2 \pm 31.1\%$ and $\Delta V_{sustained} = 64.8 \pm 9.4\%$; for Rab10 CA, $\Delta V_{transient} = 97.7 \pm 16.5\%$ and $\Delta V_{sustained} = 23.0 \pm 7.1\%$; *Figure 4b, d and f*), while CA-Rab4 slightly increased the transient phase of sLTP (but not statistically significant; for control, $\Delta V_{transient} = 357.4 \pm 52.0\%$ and $\Delta V_{sustained} = 83.5 \pm 10.8\%$; for Rab4a CA, $\Delta V_{transient} = 463.1 \pm 61.1\%$ and $\Delta V_{sustained} = 112.7 \pm 18.3\%$; *Figure 4b, d and f*).

We further evaluated the effects of Rab10 deletion on structural and electrophysiological LTP using Rab10 conditional knockout mice (*Rab10*[fl/fl]) (*Vazirani et al., 2016*). For sLTP measurement, we biolistically transfected cultured organotypic hippocampal slices of *Rab10*[fl/fl] mice with tdTomato-fused Cre recombinase and mEGFP, or tdTomato and mEGFP as a control (J.-Y. *Chang et al., 2017*). Consistent with the Rab10-knockdown results, deletion of the *Rab10* gene increased sLTP in the stimulated spines of CA1 pyramidal neurons (*Figure 4g, h and i*). Furthermore, we crossed *Rab10*[fl/

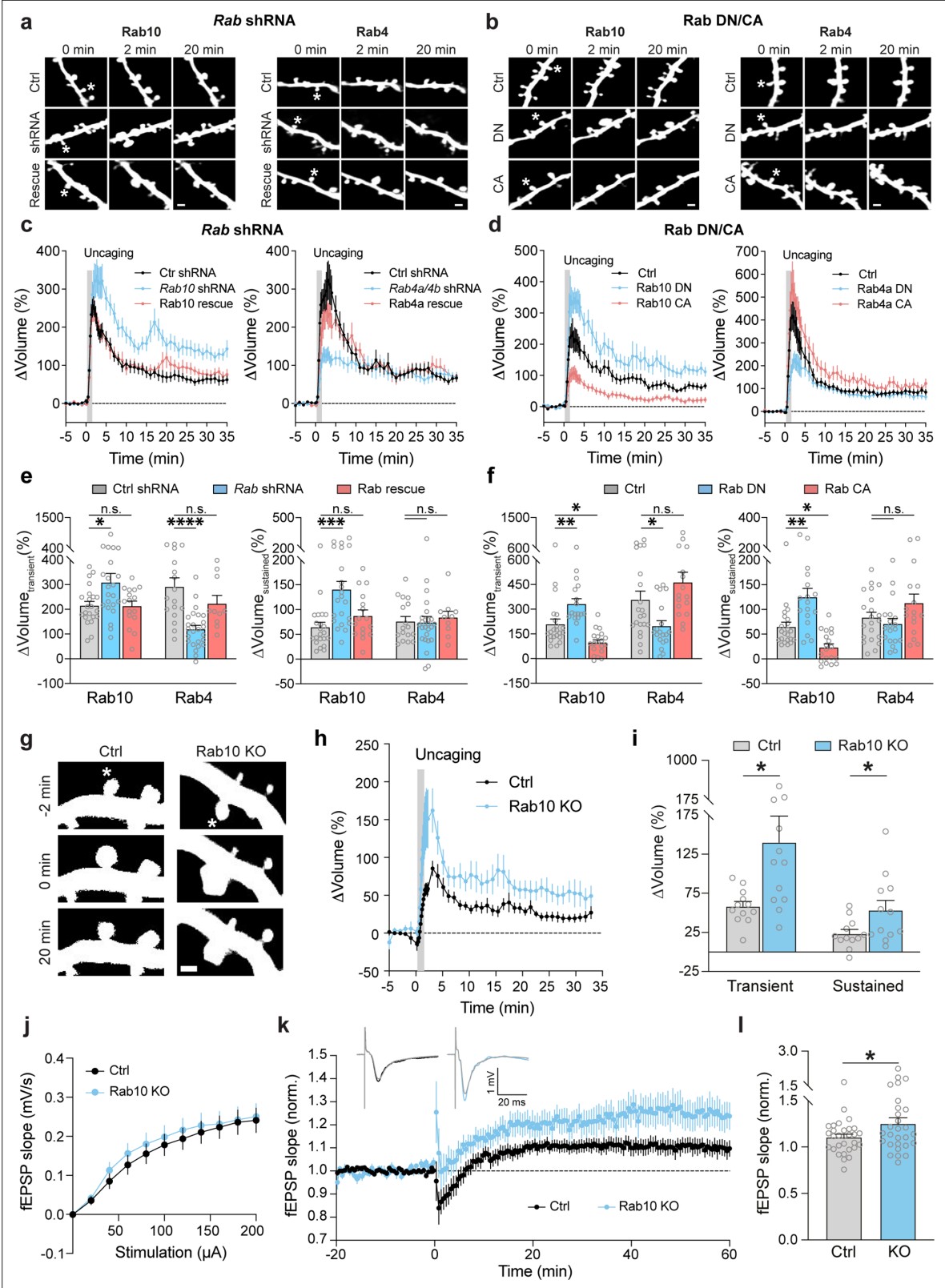

**Figure 4.** Rab4 positively regulates the transient phase of structural LTP, and Rab10 negatively regulates structural and electrophysiological LTP. (**a, b**) Representative fluorescence images of spine volume change in the stimulated spines after manipulations of Rab10 and Rab4 signaling. Asterisks indicate the stimulated spines. Scale bars represent 1 μm. (**a**) Left: rat CA1 pyramidal neurons were transfected with mEGFP and scrambled shRNA (Ctrl shRNA); mEGFP and shRNA against *Rab10* (*Rab10* shRNA); mEGFP, shRNA against *Rab10,* and mCherry-shRNA-resistant Rab10 (Rab10 rescue). Right:

*Figure 4 continued on next page*

*Figure 4 continued*

rat CA1 pyramidal neurons were transfected with mEGFP and scrambled shRNA (Ctrl shRNA); mEGFP and shRNAs against *Rab4a* and *Rab4b* (*Rab4a/4b* shRNA); mEGFP, shRNAs against *Rab4a* and *Rab4b*, and mCherry-shRNA-resistant Rab4a (Rab4a rescue). (**b**) Left: rat CA1 pyramidal neurons were transfected with mEGFP (Ctrl); mEGFP and mCherry-Rab10 [T23N] (Rab10 DN); mEGFP and mCherry-Rab10 [Q68L] (Rab10 CA). Right: rat CA1 pyramidal neurons were transfected with mEGFP (Ctrl); mEGFP and mCherry-Rab4a [S27N] (Rab4a DN); mEGFP and mCherry-Rab4a [Q72L] (Rab4a CA). (**c, d**) Averaged time course of spine volume change for experiments in **a** and **b**. Fluorescence intensity of mEGFP was used to measure the spine volume change. Data represent means ± SEM. All experiments were paired with the same day controls from the same batch of slices. (**c**) Left: N=22/20, 22/20 and 16/13 (spine/neuron) for Ctrl shRNA (black), *Rab10* shRNA (blue) and Rab10 rescue (red), respectively. Right: N=17/17, 25/21 and 10/8 (spine/neuron) for Ctrl shRNA (black), *Rab4a/4b* shRNA (blue) and Rab4a rescue (red), respectively. (**d**) Left: N=22/18, 17/15 and 19/14 (spine/neuron) for Ctrl (black), Rab10 DN (blue) and Rab10 CA (red), respectively. Right: N=21/15, 19/18, and 16/11 (spine/neuron) for Ctrl (black), Rab4a DN (blue), and Rab4a CA (red), respectively. (**e, f**) Quantitative analysis of the transient volume change (volume change averaged over 1.3–4 min, left) and the sustained volume change (volume change averaged over 20–35 min, right) for **c** in **e** and for d in **f**. Data represent means ± SEM. One-way ANOVA followed by Bonferroni's multiple comparison tests was performed (n.s., not significant, * $p<0.05$, ** $p<0.01$, *** $p<0.001$, **** $p<0.0001$). (**g**) Representative fluorescence images of stimulated spine volume change in CA1 pyramidal neurons from *Rab10*fl/fl mice transfected with tdTomato-Cre and mEGFP (Rab10 KO), or tdTomato and mEGFP as a control (Ctrl). Asterisks indicate the stimulated spines. Scale bars represent 1 μm. (**h**) Averaged time courses of stimulated spine volume change for experiments in **g**. Data represent means ± SEM. N=12 for Ctrl (black) and 12 for Rab10 KO (blue). (**i**) Quantification of the transient volume change (left) and the sustained volume change (right) for **h**. Data represent means ± SEM. Unpaired two-tailed Student's t-tests were used (* $P<0.05$). (**j**) Electrophysiological recordings showing average input/output curve of *Rab10*fl/fl:*Camk2a-Cre*+/-mice (Rab10 KO, blue, n=29) and littermate *Rab10*fl/fl:*Camk2a-Cre*-/- control mice (Ctrl, black, n=29). Data are mean ± SEM. (**k**) Time course of extracellularly recorded excitatory postsynaptic potential (fEPSP) slope before and after LTP induction in CA1 pyramidal neurons from *Rab10*fl/fl:*Camk2a-Cre*+/-mice (blue, number of slices/animals = 28/7) and littermate *Rab10*fl/fl:*Camk2a-Cre*-/- mice (black, n=30/8). Data are represented as mean ± SEM. Insets (top) are the representative traces of fEPSP before and after stimulation for Cre+ (blue) and Cre- (black) mice. (**l**) Quantification of the average fEPSP slope at 40–60 min for experiments in **k**. Data are mean ± SEM. Unpaired two-tailed Student's t-tests were used (* $p<0.05$).

The online version of this article includes the following source data and figure supplement(s) for figure 4:

**Figure supplement 1.** Validation of *Rab GTPase* shRNA and shRNA-resistant Rab GTPases, and effects of Rab knockdown on spine size and density.

**Figure supplement 1—source data 1.** Original gel photos.

**Figure supplement 1—source data 2.** Original gel photos with annotations.

**Figure supplement 2.** Deletion of Rab10 has no effect on CA3-CA1 synaptic transmission in the hippocampus.

fl mice with *Camk2a-Cre* mice (*Rab10*fl/fl:*Camk2a-Cre*+/-) to postnatally remove Rab10 from forebrain excitatory neurons (*Tsien et al., 1996*). These animals showed enhanced LTP upon theta burst stimulation (TBS) at Schaeffer collateral synapses (*Figure 4k and l*), with the basal synaptic transmission unchanged (*Figure 4j*). Control animals (*Rab10*fl/fl without Cre) showed a modest LTP under this condition (*Figure 4k and i*; *Gong et al., 2009*; *Grover et al., 2009*; *Capocchi et al., 1992*). Moreover, we monitored Schaffer collateral synaptic transmission in these mice and found no difference in the amplitude of AMPAR- and NMDAR-EPSCs, or AMPAR/NMDAR EPSC ratio (*Figure 4—figure supplement 2*). These results indicate that Rab10 is a negative regulator for electrophysiological LTP.

Overall, our results demonstrate that Rab10 negatively regulates both the transient and sustained phase of sLTP, while Rab4 positively regulates the transient phase of sLTP. These functions are consistent with the direction and time window of their activity changes during sLTP. Moreover, Rab10 negatively modulates electrophysiological LTP.

## Rab10 and Rab4 oppositely regulate activity-dependent SEP-GluA1 exocytosis during sLTP

We further examined whether Rab10 and Rab4 play roles in the exocytosis of GluA1-containing vesicles during sLTP (*Patterson et al., 2010*). To visualize newly exocytosed AMPARs from the intracellular compartments during sLTP, we combined fluorescence recovery after photobleaching (FRAP) with two-photon glutamate uncaging and two-photon imaging (*Makino and Malinow, 2009*; *Patterson et al., 2010*). Organotypic hippocampal slices were biolistically transfected with N-terminal super-ecliptic pHluorin (SEP)-tagged GluA1, mCherry, and scrambled shRNA. CA1 pyramidal neurons expressing mCherry and SEP-GluA1 were imaged under two-photon microscopy. Since SEP-GluA1 is quenched in the acidic environment of endosomes, only the population on the surface emits fluorescence (*Miesenböck et al., 1998*; *Makino and Malinow, 2009*; *Patterson et al., 2010*).

We pre-bleached surface SEP-GluA1 in a whole secondary dendrite with two-photon excitation and measured the fluorescence recovery due to exocytosis in the spines after the induction of sLTP.

Upon glutamate uncaging, the volume of the stimulated spines, measured with mCherry fluorescence, was increased by 352.8 ± 34.3% at 2 min (*Figure 5a and b*). Meanwhile, the fluorescence intensity of SEP-GluA1 was rapidly recovered from 15.4 ± 1.6% (0 min) to 108.8 ± 10.8% (2 min) in the stimulated spines (*Figure 5a and b*). However, in the non-stimulated adjacent spines and dendrites, SEP-GluA1 recovery was smaller and slower (for adjacent spines, from 17.3 ± 1.3% at 0 min to 28.9 ± 2.7% at 2 min; for dendrites, from 27.8 ± 1.4% at 0 min to 43.9 ± 1.9% at 2 min; *Figure 5a and b*). Overexpression of tetanus toxin light chain (TeTxLC), which cleaves vesicle-associated membrane protein (VAMP; *Link et al., 1992*), significantly decreased both the spine enlargement and the SEP-GluA1 recovery in the stimulated spines, suggesting that the fluorescence recovery requires exocytosis (*Figure 5c-f*, *Figure 5—figure supplement 1a*).

To investigate whether Rab10 and Rab4 are involved in this activity-dependent postsynaptic AMPAR exocytosis, we knocked down endogenous Rab10 or Rab4 by respective shRNA and monitored SEP-GluA1 exocytosis and sLTP in the stimulated and adjacent spines. No significant difference was seen in the adjacent spines or dendrites in either SEP-GluA1 recovery or mCherry intensity change among all groups (*Figure 5—figure supplement 2a–h*). Compared with scrambled shRNA, expression of *Rab10* shRNA enhanced the spine enlargement as well as SEP-GluA1 incorporation (*Figure 5a,c-f*, *Figure 5—figure supplement 1d*; measured at 2 min). These phenotypes were rescued by coexpressing shRNA-resistant Rab10 (*Figure 5a,c-f*, *Figure 5—figure supplement 1e*; measured at 2 min). Therefore, Rab10 negatively regulates GluA1 exocytosis in the stimulated spines during sLTP. In contrast, the expression of *Rab4a* and *Rab4b* shRNAs impaired spine enlargement (*Figure 5a,e,f*, *Figure 5—figure supplement 1b*; measured at 2 min). It also significantly attenuated SEP-GluA1 recovery in the stimulated spines (*Figure 5a,c,d*, *Figure 5—figure supplement 1b*; measured at 2 min). These phenotypes were rescued by coexpressing shRNA-resistant Rab4a (*Figure 5a,c-f*, *Figure 5—figure supplement 1c*; measured at 2 min). These findings suggest that Rab4 is required for GluA1 exocytosis in the stimulated spines during sLTP. Moreover, compensating for the effect of SEP-GluA1's lateral diffusion by subtracting the change in the spine surface area ($\Delta$Volume$^{2/3}$) (*Figure 5—figure supplement 2i*; *Patterson et al., 2010*) did not alter our findings. Therefore, during sLTP, Rab10 limits and Rab4 enhances SEP-GluA1 incorporation in the stimulated spines.

## Discussion

The development of novel FRET-based biosensors for Rab proteins has revealed how Rab signaling pathways regulate sLTP in single dendritic spines. In brief, NMDAR activation triggers Ca$^{2+}$ influx (~ms) and CaMKII activation (~s) (*Noguchi et al., 2005*; *Lee et al., 2009*), leading to the persistent inactivation of Rab10 (~min) and transient activation of Rab4 (~min) (*Figure 5g*). Consistent with the direction and the duration of its activity change, Rab10 negatively regulates both the transient and sustained phase of sLTP and electrophysiological LTP. In contrast, Rab4 positively regulates the transient phase of sLTP. Thus, the temporal dynamics of Rab10 and Rab4 mirror the time courses of their functions in sLTP. Furthermore, Rab10 inhibits activity-dependent AMPAR trafficking during sLTP, while Rab4 promotes this process (*Figure 5g*). These results suggest that Rab4 and Rab10 play critical roles in two membrane trafficking events – AMPAR trafficking and spine enlargement – during sLTP.

Understanding the kinetics of Rab4 and Rab10 sensors is essential for interpreting their actual activity during sLTP. The Rab4 sensor exhibits a rapid rise and fall in activation (*Figure 3*), indicating ON/OFF times of less than a few minutes. In contrast, the Rab10 sensor rapidly dissociates during sLTP induction (*Figure 2*), with OFF kinetics occurring within one minute and fast ON kinetics in response to NMDA (*Figure 1j*). Given these rapid kinetics, the observed sustained inactivation of Rab10 likely reflects its true behavior rather than sensor dynamics.

Deletion or inhibition of Rab10 enhanced spine enlargement during sLTP (*Figure 4a–i*), while inhibition of Rab10 did not alter spine size in the basal state (*Figure 4—figure supplement 1h*). This lack of change in basal spine size may be attributed to cumulative activity-dependent plasticity over time. For instance, homeostatic plasticity could have normalized basal spine size after an extended period (*Turrigiano and Nelson, 2004*; *Wefelmeyer et al., 2016*). Additionally, despite the increased spine density observed with *Rab10* shRNA knockdown in organotypic hippocampal slices (*Figure 4—figure supplement 1i*), we observed no change in synaptic transmission following Rab10 deletion in acute slices (*Figure 4j*, *Figure 4—figure supplement 2*). This discrepancy may reflect differences in developmental and homeostatic stages across these experimental models. Alternatively, it could arise from

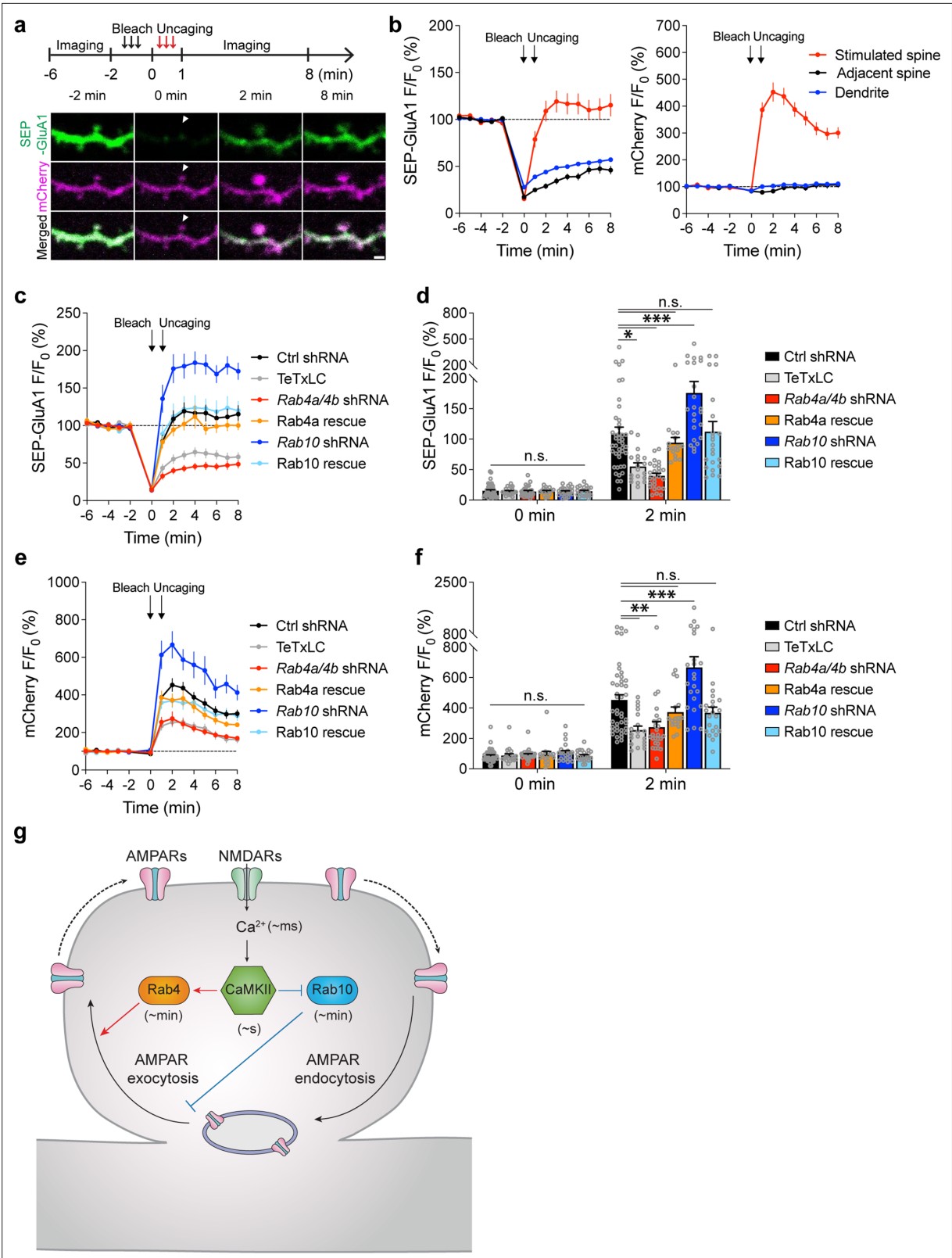

**Figure 5.** Rab4 and Rab10 positively and negatively regulate activity-dependent SEP-GluA1 exocytosis in the stimulated spines during sLTP, respectively. (**a**) Upper panel: schematic for SEP-GluA1 FRAP and two-photon glutamate uncaging experiment. Whole dendrite photobleaching was performed from –2 min to 0 min, followed by single spine glutamate uncaging from 0 min to 1 min (0.5 Hz, 60 s). Lower panel: representative pseudo color images of SEP-GluA1 (green) FRAP after two-photon glutamate uncaging in a single spine of hippocampal CA1 pyramidal neurons coexpressing mCherry

*Figure 5 continued on next page*

*Figure 5 continued*

(magenta) and scrambled shRNA. White arrowheads indicate the stimulated spine. Scale bar represents 1 μm. (**b**) Averaged time courses of SEP-GluA1 (left) and mCherry (right) fluorescence intensity ($F/F_0$) in the stimulated spine (red), adjacent spine (black), and dendrite (blue) of neurons expressing SEP-GluA1, mCherry, and scrambled shRNA. Black arrows indicate the time points after photobleaching and glutamate uncaging, respectively. Data represent mean ± SEM. N=43/35 (spine/neuron), 43/35 (spine/neuron) and 43/35 (dendrite/neuron) for the stimulated spine, adjacent spine and dendrite, respectively. (**c**) Averaged time courses of SEP-GluA1 FRAP in the stimulated spines of neurons expressing SEP-GluA1, mCherry, and scrambled shRNA (Ctrl shRNA, black, n=43/35); SEP-GluA1, mCherry, and TeTxLC (TeTxLC, grey, n=19/12); SEP-GluA1, mCherry, and shRNAs against *Rab4a* and *Rab4b* (Rab4a/4b shRNA, red, n=26/16); SEP-GluA1, mCherry, shRNAs against *Rab4a* and *Rab4b*, and shRNA-resistant Rab4a (Rab4a rescue, orange, n=16/12); SEP-GluA1, mCherry, and shRNA against *Rab10* (Rab10 shRNA, green, n=23/15); SEP-GluA1, mCherry, shRNA against *Rab10*, and shRNA-resistant Rab10 (Rab10 rescue, blue, n=22/19). Data represent mean ± SEM. All experiments were paired with the same day controls from the same batch of slices. (**d**) Quantification of SEP-GluA1 fluorescence intensity at 0 min and 2 min in the same experiments as **c**. Data represent mean ± SEM. One-way ANOVA followed by Bonferroni's multiple comparison tests was performed (n.s., not significant, * $p<0.05$, *** $p<0.001$). (**e**) Averaged time courses of mCherry fluorescence intensity in the stimulated spines of the same neurons as **c**. Data represent mean ± SEM. (**f**) Quantification of mCherry fluorescence intensity at 0 min and 2 min in the same experiments as c. Data represent mean ± SEM. One-way ANOVA followed by Bonferroni's multiple comparison tests (n.s., not significant, ** $p<0.01$, *** $p<0.001$). Please note that the Ctrl shRNA samples in c-f are the same as those in a-b. (**g**) Proposed model for Rab4 and Rab10 mediated AMPAR trafficking and sLTP. Activation of postsynaptic NMDARs triggers $Ca^{2+}$ influx (~ms) and CaMKII activation (~s), which is relayed by the transient activation of Rab4 (~min) and persistent inactivation of Rab10 (~min). Rab4 activation and Rab10 inactivation result in the potentiated AMPAR exocytosis and sLTP induction in single dendritic spines.

The online version of this article includes the following figure supplement(s) for figure 5:

**Figure supplement 1.** Rab4 and Rab10 regulate activity-dependent GluA1 exocytosis in the stimulated spines during sLTP.

**Figure supplement 2.** Fluorescence intensity of SEP-GluA1 and mCherry in adjacent spines and dendrites.

distinct regulatory mechanisms influencing spine size and nanometer-level AMPAR trafficking (*Zito et al., 2004*; *Haas et al., 2018*).

Previous studies have suggested that gating of signaling regulates kinase-phosphatase balance during LTP. For example, activation of the cyclic adenosine monophosphate (cAMP) pathway inactivates protein phosphatase, thereby gating CaMKII signaling (*Yagishita et al., 2014*; *Genoux et al., 2002*; *Blitzer et al., 1998*). Here we demonstrated that Rab10 is inactivated during sLTP and plays inhibitory roles in AMPAR trafficking and spine enlargement, suggesting that Rab10 acts as a gate for the membrane trafficking events during sLTP. Thus, gating of inhibitory signaling is likely a common mechanism in synaptic plasticity.

On the other hand, Rab4 facilitates membrane expansion and AMPAR trafficking, specifically during the transient phase of sLTP. In a previous study, LTP onset appears to be delayed in Rab4 knock-down neurons, implicating that Rab4 is required for the rapid increase of AMPAR current during LTP induction (*Brown et al., 2007*). Indeed, Rab4 is rapidly recruited into the spines in the transient phase of sLTP (*Figure 2—figure supplement 5d–f*). Consistently, recent ultrastructural analyses also demonstrated the increase of PSD complexity and membrane expansion during the transient phase of sLTP (*Sun et al., 2021*). RhoA is another signaling molecule that plays a role specifically in the transient phase, likely by reorganizing the actin cytoskeleton in spines (*Murakoshi et al., 2011*). These signaling processes, which rapidly reorganize spine structure during the transient phases, would be critical for shaping the onset of LTP and sLTP.

Interestingly, Rab4, 5, 7, 8, and 10 sensors are all activated upon low-dose NMDA application (*Figure 1j*, *Figure 1—figure supplement 1e*). However, they displayed distinct time courses and durations of activity change (*Figure 1j*, *Figure 1—figure supplement 1e*). It is known that bath application of low-concentration NMDA induces LTD and dephosphorylation of AMPARs in the hippocampus (*Lee et al., 1998*). During LTD, the rate of AMPAR internalization outweighs the rate of AMPAR exocytosis, resulting in a reduced number of synaptic AMPARs. Since Rab proteins are localized on different endosomes and coordinate individual parts of receptor trafficking, their distinct time courses and activity durations may reflect the involvement of related endosomes in LTD. Particularly, the Rab10 sensor displayed bi-directional activity changes in response to sLTP or chemical LTD induction. To fully understand Rab10's function in synaptic plasticity, it would be necessary to investigate its involvement in LTD.

Notably, the inactivation of Rab10 is persistent and lasts for over 30 min. Despite different time courses of activity during sLTP, other small GTPases RhoA, Cdc42, Rac1, and Ras also remain activated over 20 min (*Murakoshi et al., 2011*; *Hedrick et al., 2016*; *Harvey et al., 2008*). Moreover, BDNF-TrkB signaling is rapidly activated in the spines and remains elevated for at least 60 min during

sLTP (*Harward et al., 2016*). The persistent inactivation pattern of Rab10 is possibly defined by the activity of its associated endosomal organelle in sLTP. Previous studies showed that Rab10 is localized on various membranes, including the early endosome, recycling endosome, endoplasmic reticulum (ER), Golgi, and trans-Golgi network (TGN). With the SLENDR-mediated knockin technique, we found that endogenous Rab10 is majorly overlapped with Rab7-labeled lysosome, partially colocalized with Rab11-labeled recycling endosome, but separated from Rab5-labeled early endosome in vivo (*Figure 1—figure supplement 1j*). These results implicate that Rab10 may mediate the transport from recycling endosome to lysosome for protein degradation. Therefore, inactivation of Rab10 possibly inhibits AMPAR degradation pathway, resulting in more available AMPARs for synaptic insertion and synaptic potentiation.

Pharmacological analysis of Rab10 and Rab4 signaling pathways indicated that they are downstream of CaMKII, but not ERK. Previously, the Ras-ERK pathway has been implicated in regulating AMPAR delivery and sLTP (*Harvey et al., 2008*; *Patterson et al., 2010*; *Zhu et al., 2002*). Our data suggest that Rab10 and Rab4 act as parallel pathways that are independent of Ras-ERK signaling. The steps between CaMKII activation and Rab10 inactivation or Rab4 activation are still unclear. One possibility is through the direct phosphorylation of Rab proteins (*Steger et al., 2016*; *van der Sluijs et al., 1992a*), or indirect regulations by Rab GAPs or GEFs. Interestingly, in contrast to the Ras-ERK pathway, which shows extensive spreading into dendrites or nucleus (*Yoshimura et al., 2010*; *Tang and Yasuda, 2017*; *Zhai et al., 2013*), the activity of Rab10 and Rab4 is restricted to the stimulated spines during sLTP. Thus, Rab10 and Rab4 may regulate local membrane trafficking in spines, whereas ERK may regulate AMPAR exocytosis in dendrites, through different downstream effectors. Notably, a spine-restricted pattern of signaling activity has also been observed for Cdc42 and Cofilin activation during sLTP, which promotes the actin polymerization (*Murakoshi et al., 2011*; *Bosch et al., 2014*). Given that some Rab proteins are known to regulate actin polymerization (*Lanzetti, 2007*), it is plausible that they influence AMPAR exocytosis and spine enlargement through modulation of actin dynamics. Thus, the combination of local membrane trafficking and actin polymerization in spines appears important for spine expansion during sLTP.

Finally, our results highlight the diverse roles of Rab proteins in the orchestrated regulation of membrane trafficking during sLTP. It appears that Rab proteins operate in different directions and time windows. Whereas Rab10 negatively regulates sLTP for a long time (~30 min), Rab4 positively regulates the initial ~5 min of sLTP. Coordination of the upregulation and downregulation of the activity of various Rab proteins with unique functional and temporal properties would allow for the flexible and reliable control of membrane trafficking in multiple forms of spine structural plasticity. Although we measured the activity of only two Rab proteins in sLTP, it is likely that other members among the ~60 Rab family proteins are critical for different spatiotemporal aspects of spine structural plasticity. Imaging the activity of other Rab proteins using similar Rab sensors will hopefully reveal the coordinated signaling that regulates membrane trafficking during synaptic plasticity.

## Materials availability statement

Plasmids developed in this study are available at Addgene. Further information and requests for resources and reagents should be directed to and will be fulfilled by the lead contact Ryohei Yasuda ( ryohei.yasuda@mpfi.org).

## Methods
### HEK 293T cells

HEK 293T cells (Fisher Scientific) were grown in DMEM (Gibco) supplemented with 10% fetal bovine serum (Invitrogen) and 1% penicillin-streptomycin (Invitrogen). The cell cultures were maintained at 37°C in a 5% $CO_2$ humidified atmosphere.

### Rat

CD wild-type rats (CD IGS) purchased from Charles River Laboratories were used for preparation of hippocampal organotypic slice culture and dissociated postnatal cortical neuron culture. Both male and female animals were used and randomly allocated to experimental groups. All the experiments were performed in accordance with guidelines from the US National Institutes of Health and were

approved by Duke University Medical Center and the Institutional Animal Care and Use Committee of Max Planck Florida Institute for Neuroscience.

## Mice

Rab10 conditional knockout (*Rab10^fl/fl*) mouse was a gift from Dr. Timothy E McGraw (*Vazirani et al., 2016*). *Camk2a-Cre* mice were previously reported (*Tsien et al., 1996*). Swiss Webster mice were obtained from Charles River for endogenous Rab10 knockin experiments. Both male and female mice were used. All the experiments were performed in accordance with guidelines from the US National Institutes of Health and were approved by the Institutional Animal Care and User Committee of Max Planck Florida Institute for Neuroscience.

## Organotypic slice culture

Organotypic rat or mouse hippocampal slices were prepared at postnatal day 6 or 7, as previously described (*Stoppini et al., 1991*). Briefly, coronal hippocampal slices were dissected at 400 μm thickness using a McIlwain tissue chopper (Ted Pella, Inc). The slices were cultured on hydrophilic PTFE membranes (Millicell, Millipore), which were inserted in the culture medium containing 8.4 mg/ml MEM (Sigma), 20% horse serum (Gibco), 1 mM L-Glutamine (Sigma), 5.2 mM NaHCO$_3$, 12.9 mM D-Glucose, 0.075% Ascorbic acid, 30 mM Hepes,1 μg/ml Insulin, 1 mM CaCl$_2$ and 2 mM MgSO$_4$. The slice cultures were maintained at 35 °C in a 5% CO$_2$ humidified atmosphere.

## Dissociated neuron cultur

Dissociated postnatal cortical cultures were prepared as previously published (*Shibata et al., 2015*). Briefly, cortices dissected from newborn rats (random male and female) were triturated and plated into 5 cm dishes coated with 50 μg/ml PLL (Sigma) in culture medium consisting of Basal Medium Eagle (BME) supplemented with 10% heat-inactivated fetal bovine serum (Invitrogen), 35 mM glucose (Sigma), 1 mM L-glutamine (Sigma), 100 U/ml penicillin (Sigma), and 0.1 mg/ml streptomycin (Sigma). Neuron cultures were maintained at 35°C in a 5% CO$_2$ humidified atmosphere.

## DNA constructs

To generate pCAGGS-mCherry, pCAGGS-mTurquoise2, pCAGGS-mCherry-mCherry, and pCAGGS-mVenus-mVenus, the respective fluorescence protein sequence was cloned into pCAGGS backbone from Raichu-2517KX gifted from Dr. Michiyuki Matsuda (*Kitano et al., 2008*). Rat full-length *Rab4a, Rab5a, Rab7a, Rab8a, Rab11a,* and *Rab10* were amplified by PCR from rat brain cDNA library (Dharmacon, Cat# LRN1205) and cloned into pmEGFP-C1 (*Murakoshi et al., 2011*), pCAGGS-mEGFP, pCAGGS-mCherry, and pCAGGS-mTurquoise2. All FRET donors were tagged at the amino terminus of Rab proteins. The linker between FRET donors (mEGFP or m Turquoise2) and Rab GTPases is SGLRSRG. *Rabenosyn5 [439-503], EEA1[36-126], FYCO1[963–1206], Rim2 [27-175],* and *Rim1 [20-227]* cDNAs were amplified by PCR from rat brain cDNA library and inserted into pCAGGS-mCherry-mCherry and pCAGGS-mVenus-mVenus constructs. The second mVenus has a mutation L68V, which improves bleaching time by 30% (called SYFP2; *Kremers et al., 2006*). The linkers between mCherry and RBDs are SGLRSRA for the amino terminus and GSG for the carboxy terminus. The linkers between mVenus and Rim1 [20-227] are SGLRSRG for the amino terminus and GSG for the carboxy terminus. Rab dominant negative (DN) and constitutively active (CA) mutants were generated from wild type Rab GTPases by site-directed mutagenesis and subcloned into pCAGGS-mEGFP, pCAGGS-3Flag, and pCAGGS-mCherry constructs. PIK3R1 [114-313], Rin1, RNTre (USP6NL), RabGAP5 (SGSM3), GRAB (Rab3il1), Rabin8 (Rab3ip), TBC1D30, Evi5l, and VPS39 were amplified from rat brain cDNA library and cloned into pCAGGS or pCAGGS-3HA vector. Full-length Dennd4c, TBC1D15, and Rab3gap1 were amplified by PCR from MGC mouse cDNA (Dharmacon) and subcloned into pCAGGS-3HA construct. psiCHECK-2-Sal4-wt_3'UTR was a gift from Robert Blelloch (Addgene plasmid # 31862). psiCHECK-2-Rab GTPases were generated by inserting Rab GTPases into psiCHECK-2-Sal4-wt_3'UTR by XhoI/ NotI.Tetanus toxin light chain (*Eisel et al., 1993*) was subcloned into pCAGGS-3Flag construct. SEP-GluA1 was a gift from Dr. Scott Soderling at Duke University (*Wang et al., 2008*). mTurquoise2-pBAD and mVenus-pBAD were gifts from Michael Davidson (Addgene plasmid # 54844 and # 54845). The human codon-optimized *S. pyogenes* Cas9 (SpCas9) and single guide RNA (sgRNA) expression plasmid was a gift from F. Zhang (pX330, Addgene plasmid # 42230; *Cong et al., 2013*).

## Antibodies

These antibodies were used for SDS-PAGE and immunoblotting: rabbit anti-Rab10 (1:500; Cell Signaling Technology, #8127), rabbit anti-Rab4b (1:500; Thermo Fisher Scientific, #PA5-49124), mouse anti-Rab4a (1:500; ThermoFisher Scientific, #MA5-17161) and mouse anti-β-actin (1:2000; Sigma, #A5316), HRP-conjugated goat anti-rabbit (1:5000, Bio-Rad, #170–6515) and HRP-conjugated goat anti-mouse (1:5000, Bio-Rad, #172–1011). These antibodies were used for histology: rabbit anti-HA primary antibody (1:1000, Cell Signaling Technology, #3724), Goat anti-Rabbit IgG (H+L) Secondary Antibody, Alexa Fluor 568 conjugate (1:1000, Thermo Fisher Scientific, #A-11036).

## RNA interference

For shRNA-mediated knock-down of Rab4 and Rab10, we used SHCLND-NM_009003 plasmid for Rab4a (Sigma-Aldrich, TRCN0000088975), SHCLND-NM_016154 plasmid for Rab4b (Sigma-Aldrich, TRCN0000380038), and TRC-Mm1.0 plasmid for Rab10 (Dharmacon, TRCN0000100838). The respective shRNA sequences (according to manufacturer and sequencing confirmation) are CCGGAGA TGACTCAAATCATACCATCTCGAGATGGTATGATTTGAGTCATCTTTTTTG for Rab4a, GTACCGGG GTCATCCTCTGTGGCAACAACTCGAGTTGTTGCCACAGAGGATGACCTTTTTTG for Rab4b, and TTGCCTTTCGGTACAACTCTC (mature antisense) for Rab10. For shRNA control, we used scrambled shRNA with the following sequence: CCTAAGGTTAAGTCGCCCTCGCTCGAGCGAGGGCGACTTAA CCTTAGG (Addgene plasmid # 1864). To visualize transfected neurons in sLTP experiments, mEGFP was inserted into scrambled shRNA, *Rab4a* and *Rab10* shRNA by KpnI/BamHI, and *Rab4b* shRNA by BamHI/BstEII. The mEGFP expression was driven by a separate hPGK promoter (shRNA/mEGFP). For the rescue experiments, silent mutations of three amino acids were introduced at the targeted region for Rab4a and Rab10 by site-directed mutagenesis (for shRNA-resistant Rab4a, AAAGATGACTC**C**AA **C**CA**C**ACCATA; for shRNA-resistant Rab10, GAGAGTTGT**G**CC**C**AA**G**GGCAA).

## Transfection of FRET sensors in HEK 293T cells and organotypic slice cultures

HEK 293T cells were transfected with Lipofectamine 2000 following the manufacturer's recommendations (Invitrogen), and imaged 24–48 hr after transfection. For Rab GTPase FRET sensors, the ratio of transfected FRET donor and acceptor was 1:3.

After 9–13 days in culture, organotypic hippocampal slices were transfected biolistically with gene gun (*McAllister, 2000*; Bio-Rad, pressure 200 psi) using gold beads (Bio-Rad, 1.6 μm) coated with plasmids, and imaged 3–4 days after transfection. For the Rab4 FRET sensor, pmEGFP-Rab4a and pCAGGS-mCherry-Rabenosyn5 [439-503]-mCherry (1:1, 20 μg) were expressed for 3 days. For the Rab10 FRET sensor, pCAGGS-mTurquoise2-Rab10 and pCAGGS-mVenus-Rim1 [20-227]-mVenus (1:3, 40–60 μg) were expressed for 3–4 days.

## Two-photon fluorescence lifetime imaging and two-photon glutamate uncaging

We used a custom-built two-photon fluorescence lifetime imaging microscope (2pFLIM) with two Ti:Sapphire lasers (Chameleon, Coherent) as previously described (*Murakoshi et al., 2008*; *Yasuda, 2006*). One laser was tuned to 920 nm to excite both donor for lifetime measurement and acceptor for morphology. The second laser was tuned to 720 nm for glutamate uncaging. The imaging power for two lasers was controlled independently by electro-optical modulators (Conoptics). The fluorescence was collected by an objective (60 X, 1.0 numerical aperture, Olympus), separated by a dichroic mirror (Chroma, 565 nm for mEGFP/mCherry and 505 nm for mTurquoise2/mVenus), filtered by wavelength filters (Chroma, ET520/60 M-2p for mEGFP, ET620/60 M-2p for mCherry, ET480/40 M-2p for mTurquoise2, ET535/50 M-2p for mVenus), and finally detected by two independent photoelectron multiplier tubes (PMTs). We used 1.2–1.5 mW imaging power for mEGFP/mCherry sensor, and 1.6–1.8 mW for mTurquoise2/mVenus sensor.

Two-photon fluorescence lifetime imaging in HEK 293T cells was performed in imaging solution containing 20 mM HEPES (pH 7.3), 130 mM NaCl, 2.5 mM KCl, 2 mM $MgCl_2$, 2 mM $NaHCO_3$, 1.25 mM $NaH_2PO_4$ and 25 mM D-glucose.

Two-photon lifetime imaging and glutamate uncaging in organotypic slices was performed in $Mg^{2+}$-free artificial cerebrospinal fluid (ACSF; 127 mM NaCl, 2.5 mM KCl, 1.25 mM $NaH_2PO_4$, 25 mM

NaHCO$_3$, 25 mM D-glucose, aerated with 95% O$_2$ and 5% CO$_2$) with 4 mM CaCl$_2$, 4 mM MNI-caged glutamate (Tocris) and 1 µM tetrodotoxin (TTX, Enzo). Uncaging pulses (0.5 Hz, 60 s, 4–6ms, 3.5–3.8 mW) were delivered to the back focal aperture of the objective, which was around 0.5 µm from the tip of the spine head. The adjacent spine refers to the first or second spine located next to the stimulated spine, typically positioned opposite to the stimulated spine. Additionally, the size of the adjacent spine must be sufficiently large for imaging. We used a heater controller (Warner Instruments TC-344B) to monitor the temperature at 25–27°C or 33–35°C. Images were analyzed by MATLAB (MathWorks) and ImageJ.

## Pharmacological inhibition

The control and experimental groups were derived from the same batch of samples and were imaged on the same day to ensure consistent microscope settings and imaging buffer conditions, but in an unblinded manner. For the AP5 experiment, control samples were incubated in the imaging buffer for 30 min prior to the induction of sLTP. Following the control experiments, AP5 was added to the same imaging buffer, and samples were incubated for an additional 30 min before sLTP induction. In the CN21 and U0126 experiments, scrambled peptide or U0124 (as controls) was added into the imaging buffer, and samples were incubated for 30 min prior to sLTP induction. After the control experiments, the perfusion system was thoroughly washed with imaging buffer. For the experimental groups, CN21 peptide or U0126 was then added to the imaging buffer, and samples were incubated for 30 min before sLTP induction.

## 2pFLIM data analysis

As described previously (*Harvey et al., 2008*), the fraction of donor bound to acceptor was determined by fitting a fluorescence lifetime curve that summed contributions from all pixels in the image, using a double exponential function convolved with the Gaussian pulse response function, as shown in the equation:

$$F(t) = F_0 \left[ P_D H(t, t_0, \tau_D, \tau_G) + P_{AD} H(t, t_0, \tau_{AD}, \tau_G) \right]$$

In this equation, $F_0$ is the peak fluorescence before convolution, while $P_D$ and $P_{AD}$ represent the fractions of free donor and donor bound to the acceptor, respectively. $t_0$ is the time offset, $\tau_D$ is the fluorescence lifetime of the free donor, $\tau_{AD}$ is the fluorescence lifetime of donor bound with acceptor, $\tau_G$ defines the width of the Gaussian pulse response function, and $H(t)$ is a fluorescence lifetime curve with a single exponential function convolved with the Gaussian pulse response function:

$$H\left(t, t_0, t_D, t_G\right) = \frac{1}{2} \exp\left( \frac{\tau_G^2}{2\tau_D^2} - \frac{t - t_0}{\tau_D} \right) erfc\left( \frac{\tau_G^2 - \tau_D\left(t - t_0\right)}{\sqrt{2}\tau_D\tau_G} \right)$$

in which erfc is the complementary error function. We fixed $\tau_D$ as 2.46 ns, 2.60 ns, and 4.15 ns, corresponding to the free mEGFP–Rab4a, mEGFP-Rab10, and mTurquoise2-Rab10 donors, respectively. Similarly, $\tau_{AD}$ was set to the fluorescence lifetime of the donor when bound to its acceptor, with values of 1.10 ns for the mEGFP/mCherry pair and 1.60 ns for the mTurquoise2/mVenus pair.

To create the fluorescence lifetime image, we computed the mean photon arrival time, $<t>$, for each pixel using the following equation:

$$< t >= \int tF\left(t\right) dt / \int F\left(t\right) dt$$

The mean photon arrival time is then related to the mean fluorescence lifetime, $< \tau >$, by an offset arrival time, $t_0$, which is determined through image-wide fitting:

$$< \tau >=< t > - t_0$$

For small regions of interest (ROIs) within an image, such as spines or dendrites, the binding fraction ($P_{AD}$) was calculated using the formula:

$$P_{AD} = \frac{\tau_D\left(\tau_D - < \tau >\right)}{\left(\tau_D - \tau_{AD}\right)\left(\tau_D + \tau_{AD} - < \tau >\right)}$$

## Spine volume measurement

To estimate the spine volume in neurons expressing Rab sensors, we measured the integrated fluorescence intensity of mCherry-RBD-mCherry or mVenus-RBD-mCherry in the spine, which is proportional to the spine volume (*Holtmaat et al., 2005*), and normalized it by the fluorescence intensity in the thick apical dendrite from the same neuron. We further multiplied this normalized value by the volume of the point spread function, which gives the spine volume in fL (*Nimchinsky et al., 2004*; *Harvey et al., 2008*).

For the sensor experiments, we used mCherry as a volume indicator. There is significant bleed-through of donor fluorescence into the acceptor channel (*Yasuda, 2006*). However, we anticipate that this would not have a large impact on the estimation of spine volume changes, as fluorescence changes in both the red and green channels exhibit similar time courses (*Figures 2e and 3e*, *Figure 2—figure supplements 5a and 6d*).

## Discussion on pH sensitivity of fluorescent sensors

Rab4 is located outside of the endosome (*van der Sluijs et al., 1991*), making it unlikely that intra-vesicular pH changes affect fluorescence lifetime measurements. The fluorescence lifetime of EGFP is minimally pH-dependent, as its protonated state does not fluoresce (*Kneen et al., 1998*). Additionally, mCherry is highly resistant to pH variations, with a pKa of less than 4.5 (*Shaner et al., 2004*). Therefore, we conclude that endosomal pH changes are unlikely to impact our Rab activity measurements.

## NMDA application

Rat organotypic hippocampal slices (DIV 9-DIV 13) were biolistically transfected with indicated Rab sensors. After 3–4 day expression, CA1 pyramidal neurons were imaged in the basal solution (ACSF with 2 mM $CaCl_2$, 2 mM $MgCl_2$ and 1 µM TTX) for 6 min. NMDA (Tocris) was bath-applied in the zero $Mg^{2+}$ solution (ACSF with 4 mM $CaCl_2$, 15 µM NMDA and 1 µM TTX) for 2 min, and replaced by the washout solution (ACSF with 2 mM $CaCl_2$, 2 mM $MgCl_2$, 1 µM TTX and 50 µM AP5) for 32 min.

## sLTP induction and measurement

Rat organotypic hippocampal slices were biolistically transfected with indicated constructs at days in vitro 9–13 (DIV 9-DIV 13). The constructs for Rab shRNA knockdown experiments were: scrambled shRNA/mEGFP (Ctrl shRNA); *Rab4a* shRNA/mEGFP and *Rab4b* shRNA/mEGFP (*Rab4a/4b* shRNA); *Rab4a* shRNA/mEGFP, *Rab4b* shRNA/mEGFP, and pCAGGS-mCherry-shRNA-resistant Rab4a (Rab4a rescue); *Rab10* shRNA/mEGFP (*Rab10* shRNA); *Rab10* shRNA/mEGFP and pCAGGS-mCherry-shRNA-resistant Rab10 (Rab10 rescue). The constructs for DN- and CA-Rab overexpression experiments were: pCAGGS-mEGFP (Ctrl); pCAGGS-mEGFP and pCAGGS-mCherry-Rab DN (Rab DN); pCAGGS-mEGFP and pCAGGS-mCherry-Rab CA (Rab CA). The constructs were expressed for 4–5 days for *Rab* shRNA knock-down and 2–3 days for DN- or CA-Rab mutant overexpression. CA1 pyramidal neurons were imaged in ACSF (aerated with 95% $O_2$ and 5% $CO_2$) with 4 mM $CaCl_2$, 4 mM MNI-caged glutamate and 1 µM TTX. Two-photon glutamate uncaging (0.5 Hz, 60 s, 4–6ms, 3.5–3.8 mW) was performed at a single spine. All experiments were paired with the same day controls from the same batch of slices. The acquired images were analyzed by MATLAB (MathWorks).

Organotypic hippocampal slices of *Rab10*<sup>fl/fl</sup> mice were biolistically transfected with tdTomato-Cre and mEGFP, or tdTomato and mEGFP as a control at DIV 10. At DIV 14, CA1 pyramidal neurons were imaged in ACSF (aerated with 95% $O_2$ and 5% $CO_2$) with 4 mM $CaCl_2$, 4 mM MNI-caged glutamate, and 1 µM TTX. Two-photon glutamate uncaging (0.5 Hz, 60 s) was performed at a single spine. All experiments were paired with the same day controls from the same batch of slices. The acquired images were analyzed by MATLAB (MathWorks).

## Spine size and spine density measurement

Rat organotypic hippocampal slices were biolistically transfected with indicated constructs at DIV 9. The constructs for *Rab* shRNA knockdown experiments were: scrambled shRNA/mEGFP (Ctrl shRNA), *Rab4a* shRNA/mEGFP and *Rab4b* shRNA/mEGFP (*Rab4a/4b* shRNA), or *Rab10* shRNA/mEGFP (*Rab10* shRNA). After 4 days of expression, CA1 pyramidal neurons were imaged in ACSF by two-photon microscopy at 25–27°C. All experiments were paired with the same-day controls from the same batch of slices. The acquired images were analyzed by MATLAB (MathWorks) and ImageJ.

## Activity-dependent SEP-GluA1 exocytosis

Rat organotypic hippocampal slices were biolistically transfected with indicated constructs at DIV 9-DIV 13. The DNA constructs for each condition were: pCAGGS-mCherry, SEP-GluA1, and scrambled shRNA (Ctrl shRNA); pCAGGS-mCherry, SEP-GluA1, and pCAGGS-3Flag-TeTxLC (TeTxLC); pCAGGS-mCherry, SEP-GluA1, *Rab4a* shRNA, and *Rab4b* shRNA (*Rab4a/4b* shRNA); pCAGGS-mCherry, SEP-GluA1, and *Rab10* shRNA (*Rab10* shRNA); pCAGGS-mCherry, SEP-GluA1, *Rab4a* shRNA, *Rab4b* shRNA, and pCAGGS-3Flag-shRNA-resistant Rab4a (Rab4a rescue); pCAGGS-mCherry, SEP-GluA1, *Rab10* shRNA, and pCAGGS-3Flag-shRNA-resistant Rab10 (Rab10 rescue). After 4 days expression, CA1 pyramidal neurons were imaged in ACSF (aerated with 95% $O_2$ and 5% $CO_2$) with 4 mM $CaCl_2$, 4 mM MNI-caged glutamate, and 1 μM TTX at 25–27°C. All experiments were paired with the same day controls from the same batch of slices. After taking five baseline images of a secondary apical dendrite (1 min interval) with 1.2–1.5 mW imaging laser power, we bleached the whole dendrite by increasing the imaging laser power to 4.0–4.5 mW for 2 min. We then performed two-photon glutamate uncaging (0.5 Hz, 60 s, 4–6ms, 3.5–3.8 mW) at a single spine and collected eight images (1 min interval). The acquired images were analyzed by MATLAB (MathWorks) and ImageJ.

In *Figure 5—figure supplement 2*, we addressed the impact of SEP-GluA1's lateral diffusion by subtracting the change in spine surface area ($Volume^{2/3}$) rather than correcting for spine volume. This approach is necessary because SEP-GluA1 is only fluorescent on the cell surface, while the fluorescence intensity of cytosolic proteins such as mCherry is proportional to spine volume. Therefore, the overall fluorescence change ($\Delta F$) should be the addition of the contribution from AMPAR trafficking ($\Delta F_t$) and the change in surface area ($\Delta S$) multiplied by the remaining SEP-GluA1 fluorescence per unit area ($f$):

$$\Delta F = \Delta F_t + f\Delta S$$

Since fluorescence immediately after photobleaching (before AMPAR trafficking happens), $F_o$, is given by $fS$ ($S$ is the surface area of the spine):

$$\frac{\Delta F}{F_0} = \frac{\Delta F_t}{F_0} + \frac{f\Delta S}{fS}$$

$$= \frac{\Delta F_t}{fS} + \frac{\Delta S}{S}$$

Assuming that the surface area change ($\Delta S/S$) is the volume change ($\Delta V/V$) to the power of 2/3, the contribution of the AMPAR trafficking can be calculated as:

$$\frac{\Delta F_t}{F} = \frac{\Delta F}{F} - \left(\frac{\Delta V}{V}\right)^{2/3}$$

Thus, we obtained the contribution of AMPAR trafficking by subtracting the volume change ($\Delta V/V$) to the power of 2/3 from the fluorescence recovery ($\Delta F/F$).

## Validation of SLENDR-mediated Rab10 knockin

The SLENDR technique was as previously described (*Mikuni et al., 2016*). Briefly, Rab10 sgRNA (IDT) was ligated into the sgRNA scaffold of pX330 (*Ran et al., 2013*) to generate Rab10 sgRNA-expressing plasmid. Rab10 sgRNA-expressing plasmid and single-stranded oligodeoxynucleotides (ssODNs; IDT) for homology-directed repair (HDR) were transfected into Neuro 2 a cells by electroporation. Genomic DNA from the electroporation-transduced Neuro 2 a cells was isolated with DNeasy Blood & Tissue Kit (QIAGEN) following the manufacturer's instruction. Genomic PCR was performed using extracted DNA as a template with corresponding primer set. The PCR product was purified by QiaQuick gel extraction kit (QIAGEN) and then proceeded to Sanger sequencing to detect 2XHA insertion.

    sgRNA target sequences (5'–3')
    Rab10: ACGTCTTCTTCGCCATTGGGAGG
    Rab4a: GCGGAGCTGTGGCGGCAGAA
    ssODNs sequence (5'–3', upper case: 2XHA tag sequence)

ggagttggttgtagtgagcagttccgatcccttggggctaccggcggcgagcgcccgagccgctcctcccaatgTACCCA
TACGATGTTCCAGATTACGCTTACCCATACGATGTTCCAGATTACGCTgcgaagaagacgtacgacct
gcttttcaagctgctcctgatcggggactcgggagtgggcaagacctgcgtc
Rab10 primer set, recombination
Rab10-2xHA-Fw: GCTCCTCCCAATGTACCCAT
Rab10-RV: AGAAACCGGATTCTGGAACG
Rab10 primer set, control
Rab10-FW: TTTCAAGCTGCTCCTGATCG
Rab10-RV: AGAAACCGGATTCTGGAACG

## In utero electroporation and histology for SLENSR-mediated Rab10 knockin

In utero electroporation (IUE) was performed as previously described. Electroporation was performed at E13-E14 in anesthetized mice. For endogenous Rab10 localization, 1–2 µl mixture of plasmids (pX330-Rab10 sgRNA and pPB-CAG-mEGFP, 1 µg/µl) and ssODNs (20 µM) was injected into the lateral ventricle of each pup. For colocalization examination of endogenous Rab10 and endosomal markers, 1–2 µl mixture of plasmids (pX330-Rab10 sgRNA and pCAGGS-mEGFP-Rab5a, pCAGGS-mEGFP-Rab7a or pCAGGS-mCherry-Rab11a, 1 µg/µl) and ssODNs (20 µM) was injected into the lateral ventricle of each pup.

With ketamine-xylazine anesthesia (100 µg of ketamine –10 µg of xylazine per g of body weight, i.p.), mice were perfused with 4% paraformaldehyde in 0.1 M phosphate buffer, pH 7.4, and the brain was fixed for 4–12 hr. After washing with PBS, coronal vibratome sections (50 µm in thickness) were prepared (VT1200, Leica). For immunohistochemistry, slices were permeabilized with 0.3–0.4% Triton X-100 in PBS, blocked with 5% normal goat serum and 2% BSA or 5% normal donkey serum in PBS, and incubated overnight with rabbit anti-HA primary antibody (1:1000, Cell Signaling Technology). After 1–3 hr incubation with Alexa Fluor-conjugated secondary antibodies (Invitrogen) followed by DAPI staining (0.1 µg/ml, Life Technologies), slices were imaged using a confocal laser-scanning microscope (LSM880 with Airyscan, Zeiss). Images were processed and analyzed using the Zen (Zeiss) and the Fiji software.

## Dual-luciferase reporter assay

HEK 293T cells were plated into 24-well plates and cotransfected with psiCHECK-2-Rab GTPases and the respective shRNA at a 1:3 ratio. For positive control, we used shRNA against *hRluc* with the following sequence: TCATAGTAGTTGATGAAGGAG (mature antisense). After 48 hr transfection, luciferase activity was measured using the Dual-Luciferase Reporter Assay System (Promega) following the manufacturer's protocol. After removing the culture medium, cells were briefly rinsed in pre-warmed 0.1 M phosphate-buffered saline (PBS) and lysed in 100 µL of 1 X passive lysis buffer in the luciferase assay kit. After gently shaking for 15 min at room temperature, samples were prepared in a 96-well plate for luminescence measurement using the GloMax-Multi Detection System (Promega). For data analysis, the hRluc (firefly luciferase) luminescence was normalized by the hluc+ (*Renilla* Luciferase) luminescence in each well to control for transfection efficiency. All experiments were paired with the same day controls from the same batch of HEK 293T cells.

## Lentivirus infection in dissociated neuron cultures

Dissociated postnatal cortical cultures were prepared as previously published (*Shibata et al., 2015*). Both male and female animals were used and randomly allocated to experimental groups. Proliferation of non-neuronal cells was inhibited by adding Cytosine arabinoside (2.5 µM) at DIV 2. At DIV 6, cultures were infected with lentiviral particles containing *Rab4a* shRNA/mEGFP (*Rab4a* shRNA); *Rab4b* shRNA/mEGFP (*Rab4b* shRNA); *Rab4a* shRNA/mEGFP and *Rab4b* shRNA/mEGFP (*Rab4a/4b* shRNA); *Rab10* shRNA/mEGFP (*Rab10* shRNA) or scrambled shRNA/mEGFP (Ctrl shRNA). At DIV 17, cells were washed with ice-cold PBS and immediately extracted with ice-cold T-PER protein extraction buffer (Pierce) supplemented with inhibitors for proteases and phosphatases (Roche). The lysates were centrifuged at 15,000 × *g* for 15 min at 4 °C and the supernatants were used for further analysis.

## SDS-PAGE and immunoblotting

Samples were prepared for standard SDS-PAGE and separated on 12% acrylamide gel (Mini-PROTEAN TGX precast gels, Bio-Rad), then transferred onto 0.2 μm pore size PVDF membranes (Millipore) using semi-dry immunoblotting (transfer buffer containing 25 mM Tris, 200 mM glycine and 20% methanol). Membranes were blocked with 5% nonfat milk (Great Value) in TBS-T (Tris Buffered Saline with 0.1% Tween-20) for 1 hr at room temperature, then incubated overnight at 4°C with primary antibodies diluted in 5% BSA in TBS-T. We used the following commercially available antibodies: rabbit anti-Rab10 (1:500; Cell Signaling Technology), rabbit anti-Rab4b (1:500; Thermo Fisher Scientific), mouse anti-Rab4a (1:500; Thermo Fisher Scientific), and mouse anti-β-actin (1:2000; Sigma). Membranes were washed three times for 15 min in TBS-T, followed by incubation for 2 hr at room temperature with HRP-conjugated goat anti-rabbit or goat anti-mouse secondary antibodies (Bio-Rad), diluted 1:5000 in 5% nonfat milk in TBS-T. Membranes were washed 3 times for 15 minutes in TBS-T, then incubated with Pierce ECL Plus western blotting substrate (for Rab10, Rab4a, and Rab4b) or Pierce ECL western blotting substrate (for β-actin) for detection of western blotted proteins. We used the Image Quant LAS4000 Imaging System (GE Healthcare) to visualize protein bands.

## Acute slice preparation

Rab 10 Cre − and Cre +littermate mice (P 30 P 50, blind-coded) were anesthetized by isoflurane inhalation and perfused intracardially with a chilled choline chloride solution. The brain was removed and placed in the same choline chloride solution composed of 124 mM Choline Chloride, 2.5 mM KCl, 26 mM NaHCO$_3$, 3.3 mM MgCl$_2$, 1.2 mM NaH$_2$PO$_4$, 10 mM Glucose, and 0.5 mM CaCl$_2$, pH 7.4 equilibrated with 95% O$_2$ and 5% CO$_2$. Coronal slices (300 μm) containing the hippocampus were cut using a vibratome (Leica) and maintained in a submerged chamber at 32 °C for 1 hr and then at room temperature in oxygenated ACSF.

## Electrophysiology

All animals were coded for their genotypes, and all recordings and analyses were conducted blindly. Slices were perfused with oxygenated ACSF containing 2 mM CaCl$_2$, 2 mM MgCl$_2$, and 100 μM picrotoxin. One glass electrode (resistance ~4 MΩ) containing the same ACSF solution was placed in the dendritic layer of the CA1 area (~100–200 μm away from the soma) while stimulating Schaffer Collateral fibers with current square pulses (0.1ms) using a concentric bipolar stimulation electrode (FHC). The initial slope of the fEPSP was monitored with custom software (Matlab). The stimulation strength was set to ~50% saturation. A 20 min stable baseline was first recorded before induction of LTP. LTP was induced by applying 5 trains of TBS. One TBS train consists of 10 bursts of 4 stimulations at 100 Hz. The inter-burst interval is 200ms and the interval between trains is 2 s. fEPSPs responses were recorded for an hour after the stimulation protocol. All data were analyzed with an in-house program written with Matlab. Data are presented as mean ± SEM.

## Whole cell recording

All animals were blinded and coded, and all recordings and analyses were conducted blindly. Animals were sedated by isoflurane inhalation and perfused intracardially with ice-cold choline chloride solution containing 124 mM choline chloride, 2.5 mM KCl, 26 mM NaHCO$_3$, 3.3 mM MgCl$_2$, 1.2 mM NaH$_2$PO$_4$, 10 mM Glucose, and 0.5 mM CaCl$_2$ (pH 7.4 equilibrated with 95% O$_2$ and 5% CO$_2$). Brains were then removed and placed in the same chilled choline chloride solution, and coronal acute slices of 300 μm from left and right hemispheres were collected and placed in oxygenated (95% O$_2$ and 5% CO$_2$) ACSF (in mM: NaCl 127, KCl 2.5, Glucose 10, NaHCO$_3$ 25, NaH$_2$PO$_4$ 1.25, MgCl$_2$ 2, CaCl$_2$ 2 mM) at 32°C for 1 hr and maintained at room temperature for the rest of the experiment. Whole-cell voltage clamp recordings of hippocampal neurons of Cre+ and Cre- Rab10 animals were performed with a Multiclamp 700B. Patch pipettes (3–6 ΩM) were filled with a Cs Methanesulfonate solution (in mM: Cs MeSO$_4$ 120, NaCl 5, TEA-Cl 10, HEPES 5, QX314-Br 5, EGTA 5, NaATP 4, MgGTP 0.3, pH 7.4). Experiments were performed at room temperature (18–20°C), and slices were perfused with oxygenated ACSF (in mM: NaCl 127, KCl 2.5, Glucose 10, NaHCO$_3$ 25, NaH$_2$PO$_4$ 1.25, MgCl$_2$ 2, CaCl$_2$ 2 mM, picrotoxin 100 μM). AMPAR and NMDAR-evoked responses were measured by voltage-clamping the cells at –70 mV and +40 mV, respectively. Postsynaptic currents were evoked by electrical stimulation using a concentric bipolar electrode (THC), with a pulse of 0.1ms. Input-output curves were first

assessed by changing the stimulation intensity from 0 to 200 µA. For the rest of the recordings, the stimulation intensity was set to the amplitude that elicited a 50% EPSC response. Recordings were filtered at 2 kHz and digitized at 10 kHz. Series and input resistances were monitored throughout the experiment. All data were acquired and analyzed with an in-house program written in Matlab. Data are presented as mean ± SEM.

## Quantification and statistical analysis

Results are reported as mean ± SEM. Statistical analysis was performed with GraphPad Prism 7 and 10. Comparisons between two groups were performed using unpaired two-tailed Student's t-tests (* $p<0.05$, ** $p<0.01$, *** $p<0.001$, **** $p<0.0001$). Comparisons for more than two groups were performed using one-way ANOVA followed by Bonferroni's multiple comparison tests or two-way ANOVA (* $p<0.05$, ** $p<0.01$, *** $p<0.001$, **** $p<0.0001$).

## Acknowledgements

This work was supported by grants from Japan Society for the Promotion of Science Overseas Research Fellowship (667 (JN)), National Institute of Health (R35NS116704(RY), R01MH080047 (RY), DP1NS096787(RY), R01MH095090 (JN and RY)) and Max Planck Florida Institute for Neuroscience. We thank the donation from Luen Fung Group (JW). We thank Drs Sridhar Raghavachari, Scott Soderling, Fan Wang, and Anne West for important discussions. We thank Dr Scott Soderling for SEP-GluA1 plasmid. We thank DrBoris Kantor and Marguerita Klein at Duke University Viral Vector Core for lentivirus production, Dr Long Yan and light microscopy facility at Max Planck Florida Institute for technical support, David Kloetzer for laboratory management and Dr Lesley Colgan for critical reading of the manuscript. We thank all other Yasuda Lab members for discussions.

## Additional information

### Competing interests

Ryohei Yasuda: A founder of Florida Lifetime Imaging, a company that sells integrated solutions for performing fluorescence lifetime imaging and FRET imaging. The other authors declare that no competing interests exist.

### Funding

| Funder | Grant reference number | Author |
| --- | --- | --- |
| Japan Society for the Promotion of Science | Overseas Research Fellowship 667 | Jun Nishiyama |
| National Institute of Neurological Disorders and Stroke | DP1NS096787 | Ryohei Yasuda |
| National Institute of Mental Health | R01MH080047 | Ryohei Yasuda |
| National Institute of Neurological Disorders and Stroke | R35NS116704 | Ryohei Yasuda |
| National Institute of Mental Health | R01MH095090 | Jun Nishiyama Ryohei Yasuda |

The funders had no role in study design, data collection and interpretation, or the decision to submit the work for publication.

### Author contributions

Jie Wang, Conceptualization, Data curation, Formal analysis, Investigation, Methodology, Writing – original draft; Jun Nishiyama, Conceptualization, Funding acquisition, Investigation, Methodology, Writing – review and editing; Paula Parra-Bueno, Data curation, Formal analysis, Investigation, Methodology, Writing – review and editing; Elwy Okaz, Data curation, Formal analysis, Investigation; Goksu

Oz, Xiaodan Liu, Tetsuya Watabe, Irena Suponitsky-Kroyter, Data curation, Formal analysis; Timothy E McGraw, Resources; Erzsebet M Szatmari, Data curation, Project administration, Writing – review and editing; Ryohei Yasuda, Conceptualization, Software, Formal analysis, Supervision, Funding acquisition, Validation, Investigation, Methodology, Project administration, Writing – review and editing

**Author ORCIDs**
Jie Wang ⓘ https://orcid.org/0000-0002-3530-5029
Jun Nishiyama ⓘ https://orcid.org/0000-0002-3627-8114
Timothy E McGraw ⓘ https://orcid.org/0000-0001-9748-263X
Erzsebet M Szatmari ⓘ https://orcid.org/0000-0002-2914-6148
Ryohei Yasuda ⓘ https://orcid.org/0000-0001-6263-9297

**Ethics**
All animal procedures were conducted in accordance with the guidelines of the U.S. National Institutes of Health and were approved by the Institutional Animal Care and Use Committee (IACUC) of the Max Planck Florida Institute for Neuroscience and Duke University Medical Center. Protocols used in this study were reviewed and approved under protocol numbers MPFI-002 and A196-19-09.

Reviewer #1 (Public review): https://doi.org/10.7554/eLife.103879.3.sa1
Reviewer #2 (Public review): https://doi.org/10.7554/eLife.103879.3.sa2
Reviewer #3 (Public review): https://doi.org/10.7554/eLife.103879.3.sa3
Author response https://doi.org/10.7554/eLife.103879.3.sa4

# Additional files

**Supplementary files**
MDAR checklist

**Data availability**
All source data of the figures and detailed plasmid information are available at: https://www.synapse.org/Synapse:syn68690323/wiki/63348. Any additional information ofon this paper is available from the lead contact upon request. Fluorescence lifetime imaging analysis software used in this study is available at (https://github.com/ryoheiyasuda/FLIMimage_Matlab_ScanImage, copy archived at *Yasuda, 2019*).

The following dataset was generated:

| Author(s) | Year | Dataset title | Dataset URL | Database and Identifier |
| --- | --- | --- | --- | --- |
| Wang J, Nishiyama J, Parra-Bueno P, Okaz E, Oz G, Liu X, Watabe T, Suponitsky-Kroyter I, McGraw TE, Szatmari EM, Yasuda R | 2025 | Rab10 inactivation promotes AMPAR trafficking and spine enlargement during long-term potentiation | https://doi.org/10.7303/syn68690323 | Synapse, 10.7303/syn68690323 |

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
