## [Editor Report · eLife Assessment]

This is an **important** study that describes the development of optical biosensors for various Rab GTPases and explores the contributions of Rab10 and Rab4 to structural and functional plasticity at hippocampal synapses during glutamate uncaging. The evidence supporting the conclusions of the paper is **solid**, and several improvements were noted by the reviewers upon revision, although some persisting inconsistencies would benefit from further clarification.

---

## [Referee Report · Reviewer #1 (Public review)]

Summary:

Wang et al. created a series of specific FLIM-FRET sensors to measure the activity of different Rab proteins in small cellular compartments. They apply the new sensors to monitor Rab activity in dendritic spines during induction of LTP. They find sustained (30 min) inactivation of Rab10 and transient (5 min) activation of Rab4 after glutamate uncaging in zero Mg. NMDAR function and CaMKII activation are required for these effects. Knock-down of Rab4 reduced spine volume change while knock-down of Rab10 boosted it and enhanced functional LTP (in KO mice). To test Rab effects on AMPA receptor exocytosis, the authors performed FRAP of fluorescently labeled GluA1 subunits in the plasma membrane. Within 2-3 min, new AMPARs appear on the surface via exocytosis. This process is accelerated by Rab10 knock-down and slowed by Rab4 knock-down. The authors conclude that CaMKII promotes AMPAR exocytosis by (i) activating Rab4, the exocytosis driver and (ii) inhibiting Rab10, possibly involved in AMPAR degradation.

Strengths:

The work is a technical tour de force, adding fundamental insights to our understanding of the crucial functions of different Rab proteins in promoting/preventing synaptic plasticity. The complexity of compartmentalized Ras signaling is poorly understood and this study makes substantial inroads. The new sensors are thoroughly characterized, seem to work very well and will be quite useful for the neuroscience community and beyond (e.g. cancer research). The use of FLIM for read-out is compelling for precise activity measurements in rapidly expanding compartments (i.e., spines during LTP). In addition to structural changes, evidence for functional LTP is provided, too.

Weaknesses:

The interpretation of the FRAP experiments (Fig. 5, Ext. Data Fig. 13) is not straightforward as spine volume and surface area greatly expand during uncaging. I appreciate the correction for added spine membrane shown in Extended Data Fig. 14i.

Pharmacological experiments were not conducted or analyzed blind, risking bias in the selection/exclusion of experiments for analysis.

---

## [Referee Report · Reviewer #2 (Public review)]

Summary:

Wang et al. developed a set of optical sensors to monitor Rab protein activity. Their investigation into Rab activity in dendritic spines during structural long-term plasticity (sLTP) revealed sustained Rab10 inactivation (>30min) and transient Rab4 activation (~5 min). Through pharmacological and genetic manipulation to constitutively activate or inhibit Rab proteins, the authors discovered that Rab10 negatively regulates sLTP and AMPA receptor trafficking, while Rab4 positively influences sLTP but only during the transient phase. These optical sensors provide new tools for studying Rab activity in cell biology and neurobiology. The distinct kinetics and functions of Rab proteins are important for understanding synaptic plasticity. However, there are some concerns regarding result inconsistencies within this manuscript and with prior work.

Strengths:

(1) The introduction of a series of novel sensors that can address numerous questions in Rab biology.

(2) The use of multiple methods to manipulate Rab proteins to reveal the roles of Rab10 and Rab4 in LTP.

(3) The discovery of Rab4 activation and Rab10 inhibition with different kinetics during sLTP, correlating with their functional roles in the transient (Rab4) and both transient and sustained (Rab10) phases of sLTP.

Weaknesses:

(1) The discrepancy between spine phenotype and sLTP potential with Rab10 perturbation remains unexplained (refer to previous Weakness #4). The basal state is the outcome of many activity-dependent processes that are physiologically relevant. It is also unclear why different preparations would yield different results. These can be experimentally addressed, and it is at least important to highlight and discuss the discrepancies.

(2) In the response, the authors estimated that the bleed-through from mEGFP-Rab is ~3% and the red channel signal from FRET changes is ~20%. The context of these percentages is unclear. Are they percentages of the total signal in the red channel, or does 3% refer to 3% of the green channel signal? Additionally, there is no explanation of how these numbers were estimated.

(3) The changes in the fEPSP slope in response to theta burst stimulation (a decrease followed by a gradual increase) differ from prior publications (e.g. PMID: 1359925, 3967730, 19144965, 20016099). The explanation of these differences due to different conditions in response to Reviewer's recommendation #6 does not seem sufficient.

---

## [Referee Report · Reviewer #3 (Public review)]

Summary:

This study examines the roles of Rab10 and Rab4 proteins in structural long-term potentiation (sLTP) and AMPA receptor (AMPAR) trafficking in hippocampal dendritic spines using various different methods and organotypic slice cultures as the biological model.

The paper shows that Rab10 inactivation enhances AMPAR insertion and dendritic spine head volume increase during sLTP, while Rab4 supports the initial stages of these processes. The key contribution of this study is identifying Rab10 inactivation as a previously unknown facilitator of AMPAR insertion and spine growth, acting as a brake on sLTP when active. Rab4 and Rab10 seems to be playing opposing roles, suggesting a somewhat coordinated mechanism that precisely controls synaptic potentiation, with Rab4 facilitating early changes and Rab10 restricting the extent and timing of synaptic strengthening.

Strengths:

The study combines multiple techniques such as FRET/FLIM imaging, pharmacology, genetic manipulations and electrophysiology to dissect the roles of Rab10 and Rab4 in sLTP. The authors developed highly sensitive FRET/FLIM-based sensors to monitor Rab protein activity in single dendritic spines. This allowed them to study the spatiotemporal dynamics of Rab10 and Rab4 activity during glutamate uncaging induced sLTP. They also developed various controls to ensure the specificity of their observations. For example, they used a false acceptor sensor to verify the specificity of the Rab10 sensor response.

This study reveals previously unknown roles for Rab10 and Rab4 in synaptic plasticity, showing their opposing functions in regulating AMPAR trafficking and spine structural plasticity during LTP.

Weaknesses:

In the first round of revision I raised these points:

(1) In sLTP, the initial volume of stimulated spines is an important determinant of induced plasticity. To address changes in initial volume and those induced by uncaging, the authors present Extended Data Figure 2. In my view, the methods of fitting, sample selection, or both may pose significant limitations for interpreting the overall results. While the initial spine size distribution for Rab10 experiments spans ~0.1-0.4 fL (with an unusually large single spine at the upper end), Rab4 spine distribution spans a broader range of ~0.1-0.9 fL. If the authors applied initial size-matched data selection or used polynomial rather than linear fitting, panels a, b, e, f, and g might display a different pattern. In that case, clustering analysis based on initial size may be necessary to enable a fair comparison between groups-not only for this figure but also for main Figures 2 and 3.

- The authors responded to this point as follows: For sensor uncaging experiments, we usually uncaged glutamate at large mushroom spines because we need to have a good signal-to-noise ratio. We just happen to choose these spines with different initial sizes for Rab4 sensor and Rab10 sensor uncaging experiments.

Even if they happen to choose these spine sizes, it is possible to compare only those that match in size. This does not require any additional experiments. Because of this, I do not find this response satisfactory.

(2) Another limitation is the absence of in vivo validation, as the experiments were performed in organotypic hippocampal slices, which may not fully replicate the complexity of synaptic plasticity in an intact brain, where excitatory and inhibitory processes occur concurrently. High concentrations of MNI-glutamate (4 mM in this study) are known to block GABAergic responses due to its antagonistic effect on GABA-A receptors, thereby precluding the study of inhibitory network activity or connectivity, which is already known to be altered in organotypic slice cultures.

- I found the Authors following response reasonable and useful:

We appreciate the reviewer's comments and would like to clarify that we have conducted experiments in acute slices for LTP using conditional Rab10 knockout (Fig. 4k, 4l), and we obtained similar results. Additionally, we have recently published findings on the behavioral deficits observed in heterozygous Rab10 knockout mice (PubMed 37156612). These studies further support our conclusions and provide additional context for our findings.

---

## [Author Response]

The following is the authors’ response to the original reviews.

**Reviewer #1 (Public review):**
Summary:Wang et al. created a series of specific FLIM-FRET sensors to measure the activity of different Rab proteins in small cellular compartments. They apply the new sensors to monitor Rab activity in dendritic spines during induction of LTP. They find sustained (30 min) inactivation of Rab10 and transient (5 min) activation of Rab4 after glutamate uncaging in zero Mg. NMDAR function and CaMKII activation are required for these effects. Knockdown of Rab4 reduced spine volume change while knockdown of Rab10 boosted it and enhanced functional LTP (in KO mice). To test Rab effects on AMPA receptor exocytosis, the authors performed FRAP of fluorescently labeled GluA1 subunits in the plasma membrane. Within 2-3 min, new AMPARs appear on the surface via exocytosis. This process is accelerated by Rab10 knock-down and slowed by Rab4 knock-down. The authors conclude that CaMKII promotes AMPAR exocytosis by (i) activating Rab4, the exocytosis driver and (ii) inhibiting Rab10, possibly involved in AMPAR degradation.Strengths:The work is a technical tour de force, adding fundamental insights to our understanding of the crucial functions of different Rab proteins in promoting/preventing synaptic plasticity. The complexity of compartmentalized Ras signaling is poorly understood and this study makes substantial inroads. The new sensors are thoroughly characterized, seem to work very well, and will be quite useful for the neuroscience community and beyond (e.g. cancer research). The use of FLIM for read-out is compelling for precise activity measurements in rapidly expanding compartments (i.e., spines during LTP).

Thank you for the evaluation.

Weaknesses:The interpretation of the FRAP experiments (Figure 5, Ext. Data Figure 13) is not straightforward as spine volume and surface area greatly expand during uncaging. I appreciate the correction for the added spine membrane shown in Extended Data Figure 14i, but shouldn't this be a correction factor (multiplication) derived from the volume increase instead of a subtraction?

We thank the reviewer for this question. The fluorescence change should reflect a subtraction of surface area, as SEP-GluA1 is only fluorescent on the cell surface, unlike cytosolic mCherry, whose fluorescence intensity is proportional to spine volume. Therefore, the overall fluorescence change (Δ*F*) should be the addition of the contribution from AMPAR trafficking (Δ*Ft*) and the change in surface area (Δ*S*) multiplied by the remaining SEP-GluA1 fluorescence per unit area (*f*):

ΔF = Δ*Ft* + *fΔS*

Since fluorescence immediately after photobleaching (before AMPAR trafficking happens), F_o_, is given by *fS* (*S* is the surface area of the spine):

*ΔF/Fo* = Δ*Ft*/ F_o_ + *fΔS* / *fS*

= Δ*Ft*/*fS* + *ΔS*/*S*

Assuming that the surface area change (*ΔS*/*S*) is the volume change (*ΔV*/*V*) to the power of 2/3, the contribution of the AMPAR trafficking can be calculated as:

Δ*Ft*/*F* = *ΔF/F* – (Δ^V/V)2/3^

This is the reason that we subtracted the contribution of the spine surface area. We have discussed this in the updated method section.

Also, experiments were not conducted or analyzed blind, risking bias in the selection/exclusion of experiments for analysis. This reduces my confidence in the results.

We acknowledge the reviewer's concern regarding the lack of blinding in our experiments. However, it is challenging to conduct blinded experiments for certain types of studies, such as sensor screening for a protein family, where we do not have expected results or a specific hypothesis prior to the experiments. In these cases, our primary readout is whether the sensor indicates any activity change upon stimulation.

To address this concern, after identifying that Rab10 is inactivated during structural LTP (sLTP) and is likely important for inhibiting spine structural LTP, we performed blinded electrophysiology experiments and obtained similar results (deletion of Rab10 from Camk2a-positive neurons leads to enhanced LTP; Fig. 4k, 4l).

**Reviewer #2 (Public review):**
Summary:Wang et al. developed a set of optical sensors to monitor Rab protein activity. Their investigation into Rab activity in dendritic spines during structural long-term plasticity (sLTP) revealed sustained Rab10 inactivation (>30min) and transient Rab4 activation (~5 min). Through pharmacological and genetic manipulation to constitutively activate or inhibit Rab proteins, they found that Rab10 negatively regulates sLTP and AMPA receptor insertion, while Rab4 positively influences sLTP but only in the transient phase. The optical sensors provide new tools for studying Rab activity in cells and neurobiology. However, a full understanding of the timing of Rab activity will require a detailed characterization of sensor kinetics.Strengths:(1) Introduction of a series of novel sensors that can address numerous questions in Rab biology.(2) Multiple methods to manipulate Rab proteins to reveal the roles of Rab10 and rab4 in LTP.(3) Discovery of Rab4 activation and Rab10 inhibition with different kinetics during sLTP, correlating with their functional roles in the transient (Rab4) and both transient and sustained (Rab10) phases of sLTP.

Thank you for the positive evaluation.

Weaknesses:(1) Lack of characterization of sensor kinetics, making it difficult to determine if the observed Rab kinetics during sLTP were due to sensor behavior or actual Rab activity.

We estimated that the kinetics of the sensors for Rab4 and Rab10 are within a few minutes. For Rab4, we observed rapid increase and decrease of the activation in response to glutamate uncaging. Thus, this would be the upper limit of the ON/OFF time constants of Rab4. For Rab10, we observed a rapid dissociation of the sensor in response to sLTP induction within ~1 min. This means that the donor and acceptor molecules are quickly dissociated during the process. Thus, the off kinetics of the sensor is within the range of minute. Meanwhile, we have the on-kinetics from Rab10 activation (donor/accepter association) in response to NMDA application and again this is within a few minutes. Given these rapid sensor kinetics in neurons, our observation of the sustained inactivation of Rab10 should reflect the true behavior of Rab10, rather than just the sensor’s response.

We revised our manuscript discussion session as follows:

“Understanding the kinetics of Rab4 and Rab10 sensors is essential for interpreting their actual activity during sLTP. The Rab4 sensor exhibits a rapid rise and fall in activation (Fig. 3), indicating ON/OFF times of less than a few minutes. In contrast, the Rab10 sensor rapidly dissociates during sLTP induction (Fig. 2), with OFF kinetics occurring within one minute and fast ON kinetics in response to NMDA (Fig. 1j). Given these rapid kinetics, the observed sustained inactivation of Rab10 likely reflects its true behavior rather than sensor dynamics.”

(2) It is crucial to assess whether the overexpression of Rab proteins as reporters, affects Rab activity and cellular structure and physiology (e.g. spine number and size).

While we did not measure the effects of Rab sensor overexpression on Rab activity or cellular structure and physiology, we showed that sLTP is similar in neurons expressing sensors. This suggests that the overexpression of Rab sensors does not significantly disrupt signaling required for sLTP.

(3) The paper does not explain the apparently different results between NMDA receptor activation and glutamate uncaging. NMDA receptor activation increased Rab10 activity, while glutamate uncaging decreased it. NMDA receptor activation resulted in sustained Rab4 activation, whereas glutamate uncaging caused only brief activation of about 5 minutes. A potential explanation, ideally supported by data, is needed.

It is a long-standing question in the field why simple NMDA receptor activation by bath application of NMDA does not induce LTP, but instead induce LTD. Rab proteins are regulated by many GEFs and GAPs and identifying different mechanisms requires completely different techniques, such as molecular screening. While our manuscript provides some insights into this question by showing that they provide opposing signals for Rab10, we believe that identifying exact mechanisms would be out of the scope of this manuscript.

(4) There is a discrepancy between spine phenotype and sLTP potential with Rab10 perturbation. Rab10 perturbation affected spine density but not size, suggesting a role in spinogenesis rather than sLTP. However, glutamate uncaging affected sLTP, and spinogenesis was not examined. Explaining the discrepancy between spine size and sLTP potential is necessary. Exploring spinogenesis with glutamate uncaging would strengthen these results. Additionally, Figure 4j shows no change in synaptic transmission with Rab10 knockout, despite an increase in spine density. An explanation, ideally supported by data, is needed for the unchanged fEPSP slope despite an increase in spine density.

We thank the reviewer for raising these important questions. In our findings, shRNA-mediated knockdown of Rab10 did not alter spine size but did increase spine density in the basal state (Extended Data Fig. 11i). This suggests that Rab10 may restrict spinogenesis without affecting spine size. Conversely, sLTP induction via glutamate uncaging is an activity-dependent process that may involve different molecular mechanisms. The signal interplay between spinogenesis and sLTP and how the exact roles of Rab signaling in different modalities of plasticity would remain elusive for the future study.

The lack of change in synaptic transmission with Rab10 knockout, despite the increase in spine density from Rab10 shRNA knockdown, may be due to different preparation and developmental stages: spine density measurements were conducted with shRNA knockdown in organotypic slices (sliced at P6-8, DIV 9-13), while electrophysiological recordings were performed in knockout mice in acute slices from adult animals (P30-60).

(5) Spine volume was imaged using acceptor fluorophores (mCherry, or mCherry/Venus) at 920nm, where the two-photon cross-section of mCherry is minimal. 920nm was also used to excite the donor fluorophore, hence the spine volume measurement based on total red channel fluorescence is the sum of minimal mCherry fluorescence from direct 920nm excitation, bleed-through from the green channel, and FRET. This confounded measurement requires correction and clarification.

We assumed that the most of fluorescence is from direct excitation of mCherry at 920 nm. The contribution from the bleed-through from mEGFP-Rab (~3%) and from FRET changes (~20%) may influence the volume measurements. However, since we observed similar fluorescence changes in the green and red channels, these factors would have only a minor impact on our results (Extended Data Fig. 6a, 6d). Also, please note that the volume change in neurons expressing sensors is just to check if the volume change is normal, and not a major point of this manuscript. We clarified this in the method section as:

“For the sensor experiments, we used mCherry as a volume indicator. We acknowledge that contributions from bleed-through from mEGFP-Rab (approximately 3%) and FRET changes (around 20%) could affect the volume measurements. However, since we observed similar fluorescence changes in both the green and red channels, we believe these factors have a minimal impact on our results (Extended Data Fig. 6a, 6d).”

**Reviewer #3 (Public review):**
Summary:This study examines the roles of Rab10 and Rab4 proteins in structural long-term potentiation (sLTP) and AMPA receptor (AMPAR) trafficking in hippocampal dendritic spines using various different methods and organotypic slice cultures as the biological model.The paper shows that Rab10 inactivation enhances AMPAR insertion and dendritic spine head volume increase during sLTP, while Rab4 supports the initial stages of these processes. The key contribution of this study is identifying Rab10 inactivation as a previously unknown facilitator of AMPAR insertion and spine growth, acting as a brake on sLTP when active. Rab4 and Rab10 seem to be playing opposing roles, suggesting a somewhat coordinated mechanism that precisely controls synaptic potentiation, with Rab4 facilitating early changes and Rab10 restricting the extent and timing of synaptic strengthening.Strengths:The study combines multiple techniques such as FRET/FLIM imaging, pharmacology, genetic manipulations, and electrophysiology to dissect the roles of Rab10 and Rab4 in sLTP. The authors developed highly sensitive FRET/FLIM-based sensors to monitor Rab protein activity in single dendritic spines. This allowed them to study the spatiotemporal dynamics of Rab10 and Rab4 activity during glutamate uncaging-induced sLTP. They also developed various controls to ensure the specificity of their observations. For example, they used a false acceptor sensor to verify the specificity of the Rab10 sensor response.This study reveals previously unknown roles for Rab10 and Rab4 in synaptic plasticity, showing their opposing functions in regulating AMPAR trafficking and spine structural plasticity during LTP.

Thank you for the positive evaluation.

Weaknesses:In sLTP, the initial volume of stimulated spines is an important determinant of induced plasticity. To address changes in initial volume and those induced by uncaging, the authors present Extended Data Figure 2. In my view, the methods of fitting, sample selection, or both may pose significant limitations for interpreting the overall results. While the initial spine size distribution for Rab10 experiments spans ~0.1-0.4 fL (with an unusually large single spine at the upper end), Rab4 spine distribution spans a broader range of ~0.1-0.9 fL. If the authors applied initial size-matched data selection or used polynomials rather than linear fitting, panels a, b, e, f, and g might display a different pattern. In that case, clustering analysis based on initial size may be necessary to enable a fair comparison between groups not only for this figure but also for main Figures 2 and 3.

We thank the reviewer for these questions. For sensor uncaging experiments, we usually uncaged glutamate at large mushroom spines because we need to have a good signal-to-noise ratio. We just happen to choose these spines with different initial sizes for Rab4 sensor and Rab10 sensor uncaging experiments.

Another limitation is the absence of in vivo validation, as the experiments were performed in organotypic hippocampal slices, which may not fully replicate the complexity of synaptic plasticity in an intact brain, where excitatory and inhibitory processes occur concurrently. High concentrations of MNI-glutamate (4 mM in this study) are known to block GABAergic responses due to its antagonistic effect on GABA-A receptors, thereby precluding the study of inhibitory network activity or connectivity [1], which is already known to be altered in organotypic slice cultures.(1) https://www.frontiersin.org/journals/neural-circuits/articles/10.3389/neuro.04.002.2009/full

We appreciate the reviewer's comments and would like to clarify that we have conducted experiments in acute slices for LTP using conditional Rab10 knockout (Fig. 4k, 4l), and we obtained similar results. Additionally, we have recently published findings on the behavioral deficits observed in heterozygous Rab10 knockout mice (PubMed 37156612). These studies further support our conclusions and provide additional context for our findings.

**Recommendations for the authors:**

**From the Senior/Reviewing Editor:**
I apologize that this took longer than intended. As you will see from the reviews there was some disagreement on several points. There was some disagreement among reviewers as to the strength of the evidence with some characterizing it as "compelling," "convincing," or "solid" while others felt the characterization of the sensors was "incomplete" and that this could have affected some of the conclusions. After extensive discussion, reviewers agreed that there was a valid concern that the conclusion that Rab10 activation is sustained could reflect a feature of the sensor. If Rab10/RBD dissociation rate were very low, and the affinity of binding were very high, this could lead to an incorrect estimate of the sustained binding due to sensor kinetics, not Rab10 activation. It was noted that this has been seen in other sensors previously (e.g. first generation PKA activity sensors), which the developers altered in later generations to increase reversibility and off kinetics of the sensor.There was also discussion of how this might be addressed and we would be interested in your comments on this issue. It was suggested that it might be helpful to revise Figure 2b to show binding fraction dynamics separately for each spine (to determine whether any actually return to baseline). Subsequently, clustering of these binding dynamics into two groups could be summarized in a version of Fig. 2e for each cluster. Differences in spine volume dynamics between these clusters would provide a measure of how strongly Rab10 binding correlates with spine volume. If they never go back to baseline, some extra experiments with longer post-plasticity induction (150mins instead of 35), might show if any reversible Rab10 binding exists post-LTP induction.An alternative suggestion was to measure the time course in the presence of a GAP or GEF, which should alter the kinetics.

Thanks for the comments. It is important that the inactivation is observed as the dissociation of the donor and acceptor of the sensor. Thus, the fact that the sensor rapidly decreases in response to uncaging means that they have rapid off kinetics. In addition, we provide evidence of a rapid increase of Rab10 in response to NMDA application, suggesting that kinetics is also rapid. We added discussion about this in the revised manuscript as:

“Understanding the kinetics of Rab4 and Rab10 sensors is essential for interpreting their actual activity during sLTP. The Rab4 sensor exhibits a rapid rise and fall in activation (Fig. 3), indicating ON/OFF times of just a few minutes. In contrast, the Rab10 sensor rapidly dissociates during sLTP induction (Fig. 2), with OFF kinetics occurring within one minute and fast ON kinetics in response to NMDA (Fig. 1j). Given these rapid kinetics, the observed sustained inactivation of Rab10 likely reflects its true behavior rather than sensor dynamics.”

There was also further discussion of the nature of the "spine volume" signal, given the fact that the two-photon cross-section of mCherry is minimal at 920nm. It was suggested that this could be due to direct acceptor excitation rather than FRET, but there was agreement that further clarity on this issue would be valuable.

We assumed that the most of fluorescence is from direct excitation of mCherry at 920 nm. The contribution from the bleed-through from mEGFP-Rab (~3%) and from FRET changes (~20%) may influence the volume measurements. However, since we observed similar fluorescence changes in the green and red channels, these factors would have only a minor impact on our results (Extended Data Fig. 6a, 6d). Also, please note that the volume change in neurons expressing sensors is just to check if the volume change is normal, and not a major point of this manuscript. We clarified this in the method section as:

“For the sensor experiments, we used mCherry as a volume indicator. We acknowledge that contributions from bleed-through from mEGFP-Rab (approximately 3%) and FRET changes (around 20%) could affect the volume measurements. However, since we observed similar fluorescence changes in both the green and red channels, we believe these factors have a minimal impact on our results (Extended Data Fig. 6a, 6d).”

The equations in the methods section differ from other papers by the same lab (e.g. Laviv et al, Neuron 2020, Tu et al. Sci Adv. 2023, Jain et al. Nature 2024). Please clarify which equations are correct.

Thanks for pointing this out. In fact, some of the equations in this manuscript were wrong, and we have corrected them in the method session.

**Reviewer #1 (Recommendations for the authors**):The effects of Rab knockdown affect both spine volume expansion and AMPAR recovery in a very similar fashion. To explain this tight coupling, the authors suggest that the availability of membrane could be a limiting factor for spine enlargement. However, some Rabs are known to affect actin dynamics, which could also explain the dual effects on AMPAR exocytosis and spine enlargement. It is not easy to come up with an experiment to differentiate between these alternative explanations, as blocking actin polymerization would likely affect exocytosis, too. The authors should consider/discuss the possibility that all of the observed Ras effects result from altered actin dynamics and that the lipid bilayer is sufficiently fluid to form a minimal surface around the expanding cytoskeleton.

Thanks for the suggestions. We included the discussion about the potential impact on the actin cytoskeleton by Rab10.

Typos: heterougenous, compartmantalization, chemaical, ballistically/biolistically (chose one).

Thanks for pointing out these typos. We have corrected them in the revised manuscript.

**Reviewer #2 (Recommendations for the authors):**
(1) Venus shows pH sensitivity, which can be significant at synapses due to pH changes. Characterizing the pH sensitivity of the sensors is essential.

Thanks for the suggestions. We did not measure pH dependence, but the PKa of these fluorophores has already been published. PKa for EGFP and Venus are both 6.0, and it is unlikely that it influenced our measurements.

(2) Presenting individual data points within all bar graphs (e.g. Fig. 2c, 2d) would enhance data transparency.

Thanks for the suggestions. We now provide individual data points in the revised main figures.

(3) In Figure 1f: Rab5 GAP expression increased the binding fraction against expectations. In addition, clarifying the color scheme in Figure 1 is needed. Are GAPs supposed to be blue/green, and GEFs red/orange? Figure 1f seems to contradict this color scheme.

Thanks for the suggestions. We clarified these issues.

(4) Quantification of the point spread function of the uncaging laser, response/settle time of the scan mirror during uncaging, and reason for changes in neighboring spines in many example images (e.g. Figure 2a, especially at 240 s; Figure 4a) would be important.

The laser is controlled by Pockels cells, which changes the laser intensity with microsecond resolution. The laser is parked for milliseconds during uncaging, much longer than the settling time of the mirror (~0.1 milliseconds). The point spread function of the uncaging laser is limited by the diffraction (~0.5 um). The uncaging spot size is mostly limited by the diffusion of uncaged glutamate, but our calcium imaging and CaMKII imaging show that the signaling is induced mostly in the stimulated spines (Lee et al., 2009; Chang et al., 2017, 2019).

(5) Please include traces for "false" sensors in stimulated spines in Figures 2b, 2e, 3b, and 3e.

The traces for the false sensors have been presented in Extended Data Fig. 3 and Extended Data Fig. 8.

(6) The traces in Figure 4k (fEPSP slope in response to theta burst stimulation, where there is a decrease in fEPSP slope followed by a gradual increase) differ from prior publications (e.g. PMID: 1359925, 3967730, 19144965, 20016099). An investigation and explanation for these differences are necessary.

We appreciate the reviewer’s comments. We performed the experiments blindly and did not try to find a condition providing control data similar to previous publications. The variations in fEPSP responses compared to prior publications may be attributed to several factors, including differences in experimental conditions such as the genetic background of the animals used, the specific protocols for theta burst stimulation, and variations in the preparation of the hippocampal slices.

(7) The title and text state that Rab10 inactivation promotes AMPAR insertion. It is unclear if this is a direct effect on AMPAR insertion or an indirect effect through membrane remodeling. Providing data to distinguish these possibilities or adjusting the title/text to reflect alternative interpretations would be beneficial.

We appreciate the reviewer's feedback. To clarify, we have revised our terminology to use "AMPAR trafficking" instead of "AMPAR insertion", as it includes both insertion and other mechanisms of AMPAR movement within the cell.

(8) Please provide an explanation for the initial Rab10 inactivation observed in Figure 1j upon NMDA application.

The application of NMDA in Fig. 1j is similar to the commonly used chemical LTD induction protocol. We used this broad stimulation approach to test whether our sensors could report Rab activity changes in neurons upon strong stimulation. However, it is an entirely different stimulation approach from the sLTP induction protocol, thus resulting in different sensor activity changes. We describe the phenomenon in the revised manuscript, but we believe that detailed analyses of Rab10 activation in response to NMDA application are beyond the scope of this manuscript.

(9) Please explain why the study focuses on Rab4 and Rab10 instead of other Rab proteins.

During our initial screening of sensors for various Rab proteins, we observed significant activity changes in the sensors for Rab4 and Rab10 upon sLTP induction. This suggested their potential relevance in synaptic processes, leading us to focus on understanding their specific roles in structural long-term potentiation.

**Reviewer #3 (Recommendations for the authors**):(1) Although it might seem trivial, the definition of adjacent spine has not been made in the text. It would be nice to have it in the Methods section.

We included it in the Methods section as follows:

"The adjacent spine refers to the first or second spine located next to the stimulated spine, typically positioned opposite the stimulated spine. Additionally, the size of the adjacent spine must be sufficiently large for imaging."

(2) The transfection method has been mentioned as "ballistic" and "biolistic" transfection. You might want to use only one term. Additionally, you can add the equipment used (Bio-rad?) and pressure (psi) in the Methods section.

We use “biolistic” throughout the manuscript now. We also added the equipment and conditions used.